# A tectonically driven Ediacaran oxygenation event

Joshua J. Williams [1,2], Benjamin J.W. Mills [3] & Timothy M. Lenton[1]

The diversification of complex animal life during the Cambrian Period (541–485.4 Ma) is thought to have been contingent on an oxygenation event sometime during ~850 to 541 Ma in the Neoproterozoic Era. Whilst abundant geochemical evidence indicates repeated intervals of ocean oxygenation during this time, the timing and magnitude of any changes in atmospheric $pO_2$ remain uncertain. Recent work indicates a large increase in the tectonic $CO_2$ degassing rate between the Neoproterozoic and Paleozoic Eras. We use a biogeochemical model to show that this increase in the total carbon and sulphur throughput of the Earth system increased the rate of organic carbon and pyrite sulphur burial and hence atmospheric $pO_2$. Modelled atmospheric $pO_2$ increases by ~50% during the Ediacaran Period (635–541 Ma), reaching ~0.25 of the present atmospheric level (PAL), broadly consistent with the estimated $pO_2 > 0.1$–0.25 PAL requirement of large, mobile and predatory animals during the Cambrian explosion.

[1] Global Systems Institute, University of Exeter, Exeter EX4 4QE, UK. [2] School of Geosciences, University of Edinburgh, Edinburgh EH8 9XP, UK. [3] School of Earth and Environment, University of Leeds, Leeds LS2 9JT, UK. Correspondence and requests for materials should be addressed to J.J.W. (email: j.j.williams-4@sms.ed.ac.uk)

The oxygenation of the Earth system was a necessary condition for the rise of complex animal life[1–5], which occurred in several steps[6–8]. The Great Oxidation Event (GOE) ~2.3 Ga saw a permanent rise in atmospheric oxygen from [9] <$10^{-5}$ PAL to[10–13] ~$10^{-4}$–$10^{-1}$ PAL, but oxygen remained well below present levels throughout the Proterozoic[12,13]. A Neoproterozoic Oxygenation Event[14] has been proposed based on a range of indirect proxies (Fig. 1). The expansion of the oceanic inventories of Mo, V and Re, and a shift towards lower $\delta^{82/76}$Se values suggest a trend towards more oxidising ocean conditions[6,15–19] across the Neoproterozoic–Phanerozoic transition, and cerium anomalies[20] point to at least a well-oxygenated shallow ocean environment after 551 Ma[21] (Fig. 1b). However, the redox state of the ocean clearly fluctuated, with a series of transient oxygenation events (Fig. 1c) getting somewhat more frequent through the Ediacaran and early-mid Cambrian[16]. These include partial and temporary oxygenation of deeper waters following the Sturtian[22] ~660 Ma, Marinoan[23] ~635 Ma and Gaskiers[24,25] ~580 Ma glaciations.

Several hypotheses have been put forward to explain a Neoproterozoic oxygenation, in which the observed ocean oxygenation trends are the result of a broader increase in atmospheric $O_2$. Following the Snowball Earth glaciations, pulses of nutrients are suggested to have entered the oceans, enhancing primary production and burial of organic matter, releasing oxygen to the atmosphere[24,26,27], but this would only have temporarily increased $pO_2$[28] (returning to the previous state after the pulse subsided). Alternatively, a sustained increase in terrestrial chemical weathering[29], potentially amplified through selective weathering of phosphorus by early terrestrial ecosystems[30] and the presence of P-rich large igneous provinces[31] could have increased $pO_2$. The expansion of an early land biosphere could also have restricted the oxidative weathering of reduced crustal rock[32], reducing the major sink of $O_2$. These mechanisms can increase atmospheric $O_2$ concentrations, but all also imply a rise in the average $\delta^{13}$C of carbonates, either by increasing organic carbon burial relative to carbonates or by restricting organic carbon weathering, and such a rise is not observed in the geological record at the time.

Here we explore an alternative mechanism of oxygenation, where changes in plate tectonics cause a rise in $pO_2$ during the late Neoproterozoic. An increased fraction of young zircon grains[33] at the Proterozoic-Phanerozoic transition points to increased continental arc volcanism, consistent with an increase in atmospheric $CO_2$ and warming of the planet from Cryogenian extreme glaciations to ≥20 °C[34] during the Cambrian. Continental volcanic arc extent varies through time, broadly matching the detrital zircon age data[35]. Moreover, assuming that global arc magmatism is proportional to subduction, mantle depletion curves also point to a maximum in subduction during the early Phanerozoic[36], supported by reconstructions of subduction-zone lengths from both plate-tectonic reconstructions[37] and kinematic modelling[38]. Whilst these methods all carry uncertainties, the geologic data all point to a general step-increase in subduction-driven degassing across the Proterozoic-Phanerozoic transition.

As there is no direct geochemical proxy for atmospheric oxygen levels, estimates largely rely on modelling approaches[39–41], based on our understanding of the long-term carbon cycle (Fig. 2). For oxygen to build up in the ocean-atmosphere system, photosynthetically-produced organic carbon must be shielded from being re-oxidised through the burial of organic carbon in seafloor sediments. Inorganic carbonates are also buried in sediments following precipitation from seawater, and both buried carbon species are returned to the ocean and atmosphere through uplift and weathering, as well as via metamorphism and degassing. Any change to the burial and return fluxes of organic carbon will cause a change in atmospheric oxygen concentration.

The transformation of $CO_2$ to organic carbon via photosynthesis imparts a significant isotopic fractionation. Buried organic carbon contains a reduced fraction of $^{13}$C atoms, leaving an excess in the ocean and atmosphere. Through this mechanism, changes in the rate of burial and weathering of organic carbon relative to the burial and weathering of carbonates alter the $\delta^{13}$C value of seawater, which is recorded in sedimentary carbonates. This effect is dampened at low $pO_2$ by the rapid adjustment of oxidative weathering rates to match carbon burial rates[12] (through changes in $O_2$), but not completely nullified unless oxidative weathering is the only $O_2$ sink. Whilst a step change in

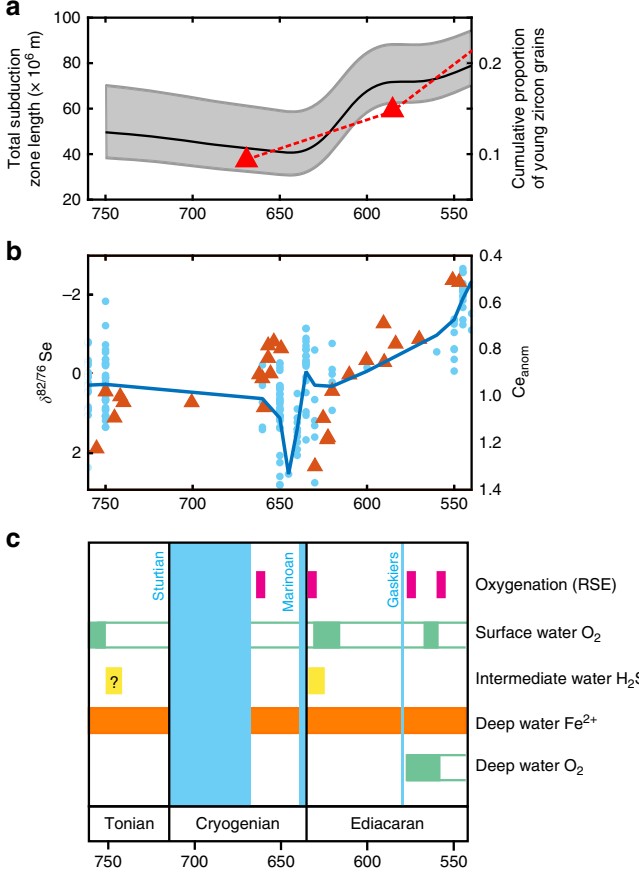

**Fig. 1** Tectonic and geochemical evidence for an Ediacaran oxygenation. **a** Total global subduction-zone length as a proxy for $CO_2$ input rate, derived from the PALEOMAP project[37]. Error estimations in grey based on observations, timing of collisions and relative plate motions (see Mills et al.[37] Methods). Also shown are the cumulative proportion of young zircon grains (red triangles), indicative of continental arc environments[33]. **b, c** Proxy compilations for the oxygenation state of the Neoproterozoic ocean: **b** Less positive selenium isotope ratios recorded in marine shales[15] ($\delta^{82/76}$Se; red triangles) indicate more oxic conditions in the global ocean. Stronger negative cerium anomalies recorded in marine cements[20] ($Ce_{anom}$; pale blue dots), defined as $Ce_{anom} < 1$, are indicative of more oxygenated regional to basin-scale conditions (the blue line is a fit through the $Ce_{anom}$ data). Both proxies suggest a trend towards more oxygenated ocean conditions during the Ediacaran period. **c** A summary of redox sensitive element (RSE; magenta) enrichment data (Mo, U, Re, V, Cr) from black shales, indicating intervals of widespread ocean oxygenation[16,17,22,23]. Below this is a summary of iron-speciation proxy data[24,25], which records localised redox conditions, hence is divided by depth range. The intervals of the Sturtian, Marinoan and Gaskiers glaciations are also indicated. These redox proxies suggest a series of transient oxygenation events during the Cryogenian and Ediacaran periods

carbonate $\delta^{13}C$ likely occurred between the mid-Proterozoic and Phanerozoic baselines[42], the later Neoproterozoic record does not show the sustained stepwise increase that would be expected if the long-term burial and weathering fluxes of organic carbon were increased relative to those of inorganic carbon. Least-squares regression analysis of carbonate carbon isotope data shows a gradual negative trend over the whole Neoproterozoic (decline of ~1–2‰), and no significant trend during the Ediacaran period where evidence for deep ocean oxygenation is found (see Supplementary Fig. 1).

Increasing the long-term tectonic input rate of $CO_2$ must result in an increase in the overall carbon burial rate to maintain steady state of the ocean-atmosphere system. If some of this carbon is buried organically then $O_2$ can rise, and if this additional burial follows the same distribution between organic carbon and carbonates as in the initial unperturbed system, then the oxygen concentration of the atmosphere can rise without any observable change in $\delta^{13}C$. This effect should be seen in carbon cycle models such as COPSE[40] and GEOCARBSULF[41], although it has not been noted in any previous analyses using these models. In addition, a similar effect may have operated over the whole of Earth history, wherein cumulative mantle $CO_2$ input can drive long-term burial of organic carbon and planetary oxygenation[7], but the model of this mechanism could not account for the timing or magnitude of any second oxygen rise after the GOE.

Here we investigate the impacts of the inferred increase in tectonic $CO_2$ input between the late Neoproterozoic and early Phanerozoic[33,35,37] by extending the COPSE Reloaded biogeochemical model[43] (see Methods) to the Ediacaran. Our model predicts a ~50% increase in atmospheric $O_2$ during the Ediacaran. The predicted magnitude and temporal dynamics of changes in redox and $^{87}Sr/^{86}Sr$ compare well to available data, as does the lack of a secular trend in carbonate $\delta^{13}C$ or $\delta^{34}S$.

## Results

**Steady-state computations.** Steady-state computations carried out at the Precambrian-Cambrian boundary (Fig. 3) show that the atmospheric oxygen reservoir size in COPSE is indeed strongly influenced by the relative rate of degassing, with higher

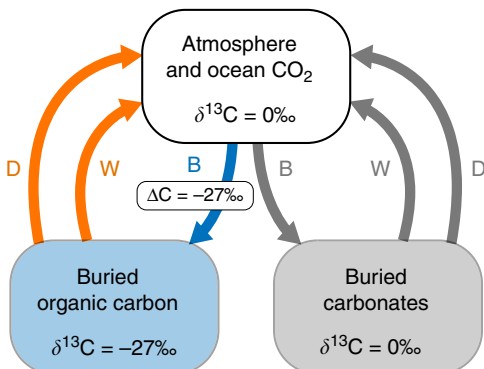

**Fig. 2** The long-term carbon cycle. The cycle is composed of fluxes between atmospheric and oceanic carbon, organic carbon and carbonate. Carbon is moved from the atmosphere and ocean to the crust through burial (B), with weathering (W), degassing and metamorphism (D) offering mechanisms by which carbon is returned to the ocean/atmosphere system. Oxygen sources are displayed as blue, oxygen sinks as orange, and other processes are displayed as grey. Fractionation of buried organic carbon relative to atmosphere and ocean carbon is denoted ΔC. The $\delta^{13}C$ value of the atmosphere and ocean reservoir can be altered by changing the proportion of organic and inorganic carbon burial, or by changing fractionation effects on either organic carbon or carbonate burial

degassing rates increasing the carbon throughput of the Earth system and atmospheric $pO_2$. The burial fluxes of both organic and inorganic forms of carbon increase when degassing is increased, but the fraction of carbon burial that is organic ($f_{org}$) remains essentially constant, allowing the long term $\delta^{13}C_{carb}$ record to remain relatively stable (Fig. 3c). This demonstrates that degassing-driven oxygenation is robust to the multiple biogeochemical feedbacks that are present in the COPSE model, which includes the consideration that a fraction of subducted material is organic carbon, and thus will consume oxygen when recycled. Nevertheless, in order to test our hypothesis fully, it is important to view the whole spectrum of model outputs. The COPSE model produces estimates for $\delta^{13}C_{carb}$, $\delta^{34}S_{seawater}$ and $^{87}Sr/^{86}Sr$, and the current model baseline reproduces these records reasonably well over the Phanerozoic, providing 'ground-truthing' that the model processes are a sensible representation of global biogeochemistry[43]. If our hypothesis for degassing-driven oxygenation is reasonable, the model should be able to reproduce the key long-term trends in these isotope records over the Ediacaran period when it is subject to our proposed increase in degassing rates alongside other expected external forcings.

We first assess the model steady states with respect to changes to both degassing rate and the other major tectonic forcing in the model—the rate of uplift and erosion (Fig. 4). Uplift and erosion enhances continental weathering fluxes in the model, which increases phosphorus delivery to the ocean and stimulates organic carbon burial. Uplift also exposes fossil organic carbon to oxidative weathering, which is the major sink of $O_2$ over geological timescales. It is likely that the processes of uplift and erosion accelerated during the late Neoproterozoic in line with the collisions that formed the supercontinents Pannotia and Gondwana: reconstructed sediment abundances increase between the Ediacaran and Cambrian[44], and rising $^{87}Sr/^{86}Sr$ ratios[45] in carbonates can be attributed to the weathering of more ancient crustal material.

In line with Fig. 3, the steady states shown in Fig. 4 show that increased degassing rates result in increased rates of burial of both organic and carbonate carbon, and increase atmospheric $O_2$, whilst maintaining an almost-static $\delta^{13}C$ ratio. Increased degassing drives a small reduction in seawater sulphate $\delta^{34}S$, due to a minor decrease in the rate of burial of pyrite sulphur relative to gypsum. In the model, rates of pyrite burial are determined based on the inferred rate of net microbial sulphate reduction, which is assumed to be positively influenced by the availability of organic matter and sulphate but is restricted by expanding deep water $O_2$[40,46,47]. This formulation produces a reasonable reconstruction of seawater $\delta^{34}S$ over the Phanerozoic[43], and in the case of increasing degassing explored here, the positive effects outweigh the negative and pyrite burial increases. However, as with the carbon cycle, burial of the oxidised form of sulphur (gypsum) increases by a similar amount, resulting in only a small change in $\delta^{34}S$ isotopic fractionation. The $^{87}Sr/^{86}Sr$ ratio is decreased under an increase in degassing rates due to the increased input of less-radiogenic mantle-derived Sr.

An increase in uplift rate acts to increase the rates of burial of both organic and carbonate carbon at steady state, due to net increases in carbonate weathering-deposition and greater delivery of phosphorus. But an increase in uplift acts to slightly decrease $O_2$ overall, due to the enhanced weathering of organic carbon. Uplift and erosion act to decrease $\delta^{13}C_{carb}$ due to increased overall rates of burial of carbonates derived from carbonate weathering (as noted peviously[48]). Increases to uplift and erosion act to increase seawater $\delta^{34}S$, because both the availability of organic C and the reduction in $O_2$ favour increased rates of microbial sulphate reduction and subsequent burial of pyrite. Finally, increased erosion rates drive a substantial increase in

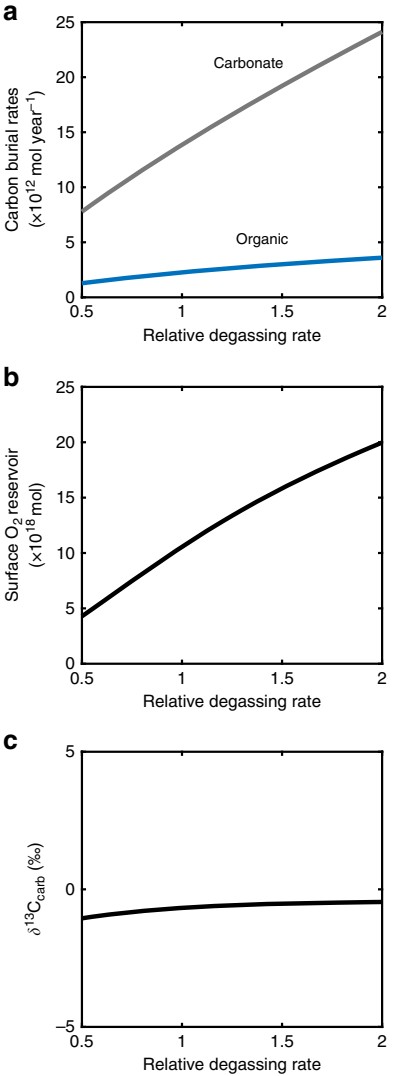

**Fig. 3** Steady-state COPSE modelling with respect to changing degassing inputs (D). **a** Organic carbon and carbonate carbon burial rates. **b** Atmosphere and ocean $O_2$ (moles), **c** $\delta^{13}C_{carb}$. All model forcings are held constant at 541 Ma, with the model allowed to stabilise for 10 Gyrs

seawater $^{87}Sr/^{86}Sr$ due to the increased weathering contributions of granites and carbonates in the model, which are relatively radiogenic.

**Monte-Carlo setup**. We now run the COPSE model forwards in time for the Ediacaran period and compare to geochemical data for $\delta^{13}C$, $\delta^{34}S$ and $^{87}Sr/^{86}Sr$. The degassing rate follows Mills et al.[37] as in Fig. 1a, and the uplift rate is set to increase linearly from 0.5 to 2 times the present day rate. The increase in uplift is chosen in order to reproduce the magnitude of the observed rise in $^{87}Sr/^{86}Sr$, but is also roughly consistent with the difference in reconstructed rates of sediment deposition between the Ediacaran and Cambrian[44]. In order to fully capture the uncertainty in model parameterisation, we run a Monte-Carlo analysis in which the COPSE model is run 10,000 times and the most important parameters are randomly sampled from an uncertainty window shown in Table 1.

We vary the assumed present-day fluxes in the model carbon cycle because these carry large uncertainties[43], and will impact the relative effects of an increase in degassing. We vary the activation

energy of seafloor weathering, as this process is a carbon sink that responds to increases in mid-ocean ridge production, thus should be increased under our assumption of increasing production and subduction rates. A highly-sensitive seafloor weathering sink would be expected to dampen the $O_2$ rise caused by an increase in degassing. Continental weathering activation energies are also varied, altering the relative strength of seafloor weathering. In the case of oxidative weathering, we test the range of possible responses to $O_2$. We also consider different values for the long-term climate sensitivity. Finally, we experiment with adding a direct mantle flux of $CO_2$ and $H_2$ into the model: COPSE does not include direct mantle input at mid-ocean ridges, and the input of reduced $H_2$ gas is an additional oxygen sink that should be amplified by increasing degassing rates. Ridge $CO_2$ input is assumed to equal $H_2$ input (in terms of mol C and mol $O_2$ equivalent) to maintain balance of the linked carbon and oxygen cycles under an additional $O_2$ sink, this value is also consistent with measurements[49]. Finally, in addition to the parameter choices, the degassing forcing is resampled every 10 Myrs from the uncertainty window of Mills et al.[37]. This allows for quite rapid changes in degassing rate, but not above the rates of change that have been suggested for more recent time periods[50].

The Monte-Carlo procedure follows that of Royer et al.[51], who sampled a wider range of parameters in a model of roughly equal complexity (GEOCARBSULF[41]), also using 10,000 runs. We choose to use a flat distribution between the parameter minimum and maximum estimates, with all values being equally likely, as we believe this is more representative of the real uncertainty than sampling from a normal distribution. The Monte-Carlo results for the carbon, oxygen and strontium cycles are shown in Fig. 5, with the dark and light shaded areas showing $\pm 0.5$ and $\pm 1$ standard deviation respectively over the whole experimental set.

**Model predictions**. Early Ediacaran $pO_2$ levels are predicted to be around ~0.2 PAL. This is consistent with recent modelling of Proterozoic oxygen regulation and with constraints on $pO_2$ from the absence of detrital pyrite[12]. It is also below upper bounds of ~0.4 PAL[47,52] or 0.5–0.7 PAL[28] inferred from ocean models, based upon the assumption of present-day nutrient levels and the presence of widespread deep ocean anoxia. Mean atmospheric oxygen is predicted to increase by around 50% during the Ediacaran period. This is primarily due to a rise in the organic carbon burial flux from ~$4 \times 10^{12}$ mol yr$^{-1}$ to ~$8 \times 10^{12}$ mol yr$^{-1}$, which is driven by increasing phosphorus input and ocean phosphate concentration. These increases ultimately stem from the increased $CO_2$ input flux through degassing and associated increase in surface temperature and weathering, which then drive carbon burial in both organic and inorganic forms. It is possible to draw a line through the model uncertainty that represents no change or even a decrease in $O_2$ levels. However, this requires moving from one edge of the uncertainty window to the other and is therefore very unlikely. Figure 6 shows a histogram of the modelled change in $pO_2$, and shows that 97% of model runs produce an increase in atmospheric $O_2$, and more than two thirds of runs produce an $O_2$ increase of between 25 and 75% of the initial value. Consequently, we infer with a high likelihood that there was a significant increase in $pO_2$ across the Ediacaran period.

Mean estimates in Fig. 5 show a slight reduction in $f_{org}$ and carbonate $\delta^{13}C$, whilst the plotted Ediacaran C isotope record shows a series of positive and negative excursions, the causes of which remain the subject of much active research e.g.,[53,54] but the long-term average remains constant (see Methods and Supplementary Fig. 1), broadly consistent with our model. Increased weathering of carbonates, and uplift-driven weathering of older

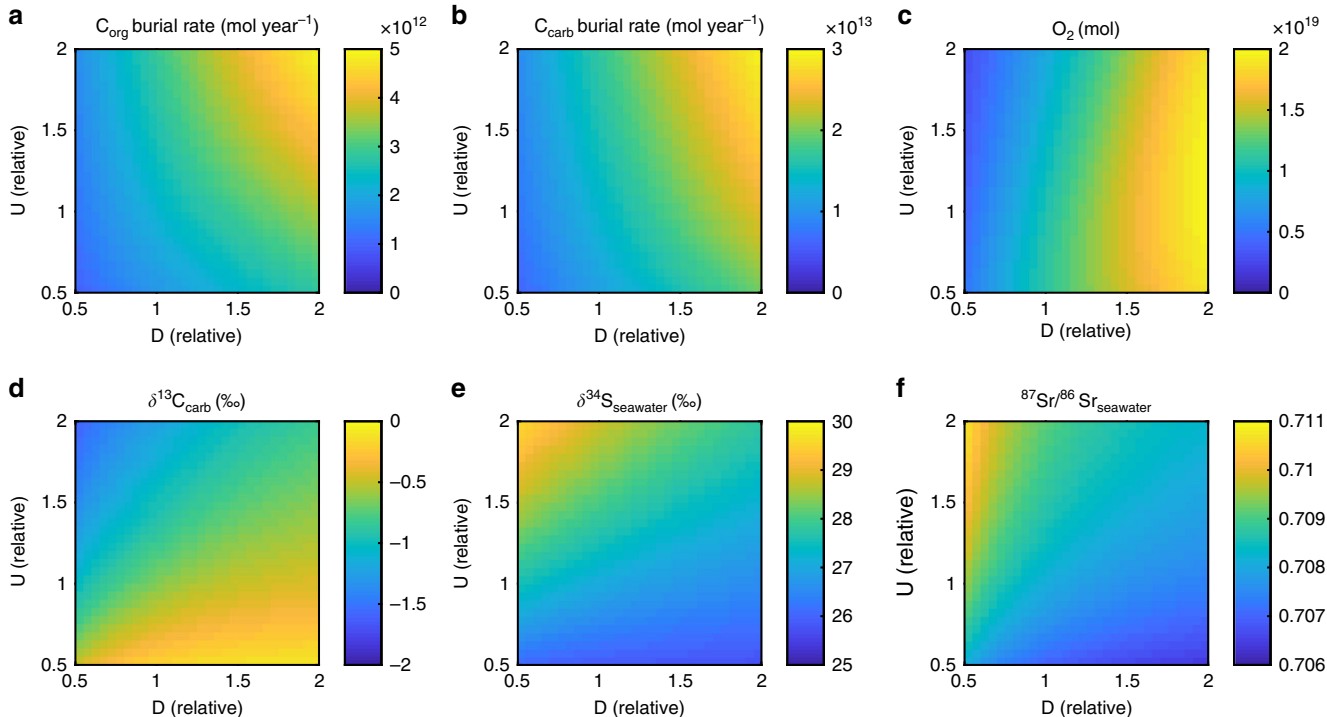

**Fig. 4** Steady states of the COPSE model with respect to changing degassing inputs (D) and global uplift/erosion rate (U). **a** Organic carbon burial rate. **b** Carbonate burial rate. **c** Atmosphere and ocean oxygen reservoir. **d** Ocean $\delta^{13}C_{carb}$. **e** Ocean $\delta^{34}S$. **f** Ocean $^{87}Sr/^{86}Sr$. All model forcings are held constant at 541 Ma, with the model allowed to stabilise for 10 Gyrs

### Table 1 Monte-Carlo parameters

| Variable | Minimum value | Maximum value | Source(s) |
|---|---|---|---|
| *Present-day values* | | | |
| Carbonate weathering | $7 \times 10^{12}$ mol/year | $14 \times 10^{12}$ mol/year | 76 |
| Total outgassing | $4 \times 10^{12}$ mol/year | $16.25 \times 10^{12}$ mol/year | 43, 77 |
| Total organic carbon burial | $5 \times 10^{12}$ mol/year | $14 \times 10^{12}$ mol/year | 43, 78 |
| Basaltic fraction of silicate weathering | 0.17 | 0.35 | 29, 76, 79 |
| Pyrite degassing | $0.12 \times 10^{12}$ mol/year | $0.38 \times 10^{12}$ mol/year | 51 |
| Gypsum degassing | $0.25 \times 10^{12}$ mol/year | $0.75 \times 10^{12}$ mol/year | 51 |
| Marine pyrite sulphur burial | $0.53 \times 10^{12}$ mol/year | $0.87 \times 10^{12}$ mol/year | 43[a] |
| Marine gypsum sulphur burial | $1 \times 10^{12}$ mol/year | $4 \times 10^{12}$ mol/year | 43[a] |
| Pyrite weathering | $0.37 \times 10^{12}$ mol/year | $0.53 \times 10^{12}$ mol/year | 43[a] |
| Gypsum weathering | $1 \times 10^{12}$ mol/year | $3 \times 10^{12}$ mol/year | 43[a] |
| Reactive phosphorus weathering | $3.7 \times 10^{10}$ mol/year | $4.7 \times 10^{10}$ mol/year | 48, 80 |
| *Weathering activation energies* | | | |
| Seafloor weathering | $40 \times 10^3$ J/mol | $100 \times 10^3$ J/mol | 81 |
| Basalt weathering | $33 \times 10^3$ J/mol | $62 \times 10^3$ J/mol | 40, 79, 82 |
| Granite Weathering | $45 \times 10^3$ J/mol | $62 \times 10^3$ J/mol | 40, 83 |
| *Reduced gas flux* | | | |
| Modern $H_2$ outgassing | 0 | $2.2 \times 10^{12}$ mol/year | 84, 85 |
| *Oxidative weathering* | | | |
| Power law dependency on $O_2$ concentration | 0 | 0.5 | 41, 43 |
| *Climate sensitivity* | | | |
| Sensitivity to a doubling of $CO_2$ | 1.5 °C | 6.0 °C | 43, 86 |

[a]denotes parameter ranges that are extended from those assessed in the COPSE paper, which only tested S cycle fluxes smaller than the baseline values

lithologies drives an increase in $^{87}Sr/^{86}Sr$ from ~0.7076 to ~0.7083, again generally consistent with the geological record—although this is expected as we have prescribed the uplift increase based partly on the Sr record. The supplementary information shows another Monte-Carlo experiment where an increase in continental uplift and erosion is not assumed over the model timeframe (See Supplementary Figs. 2–4 and Supplementary Note 1). This results in a much poorer fit to the $^{87}Sr/^{86}Sr$

record, but the results for the carbon, sulphur and oxygen cycles are very similar, and as such we conclude that our proposal of degassing-driven $O_2$ rise is robust to any expected uplift rate changes.

Figure 7 shows the sulphur cycle outputs from the Monte-Carlo experiment. With increased inputs of sulphur from continental weathering and degassing, oceanic sulphate concentration, alongside pyrite and gypsum burial rates, also increase

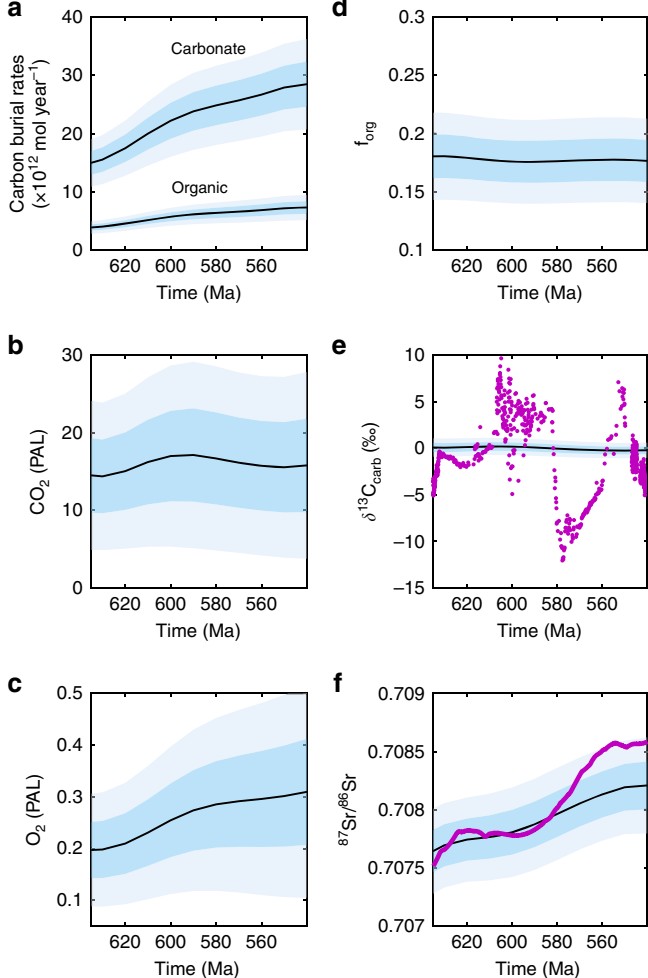

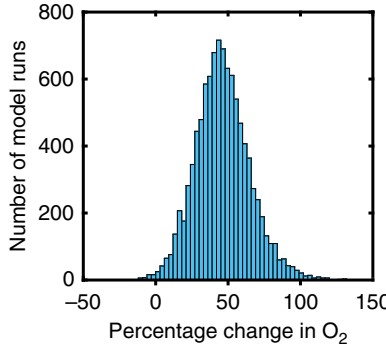

**Fig. 6** Probability distribution of surface $O_2$ reservoir change. Probability distribution of surface $O_2$ reservoir change between 640–620 Ma and 560–540 Ma for Monte-Carlo ensemble. The mean $O_2$ value is taken from each age range

**Fig. 5** Oxygen, carbon cycle and strontium isotope outputs. Monte-Carlo experiments running the COPSE model for the Ediacaran period. Degassing rates are sampled from the error window of Mills et al. (2017)[37], uplift rates increase linearly from 0.5 to 2 times present day. Values for the key model parameters controlling atmospheric oxygen are chosen for each run from the ranges shown in Table 1. Light shaded windows show ± 1 std. dev, dark shaded windows show ± 0.5 std. dev. Geologic data in purple[45,67]. **a** Carbon burial rates. **b** $CO_2$ (PAL), **c** $O_2$ (PAL), **d** $f_{org}$, **e** $\delta^{13}C_{carb}$, **f** $^{87}Sr/^{86}Sr$

under enhanced degassing. This response mimics that of the carbon cycle, where increasing input of degassed sulphur species necessitates an overall greater burial rate. Indeed, the sulphur cycle generates some of the oxygen rise in our model: pyrite burial increases by around $1 \times 10^{12}$ mol S yr$^{-1}$, generating an $O_2$ flux of $\sim 2 \times 10^{12}$ mol yr$^{-1}$, or around 30% of the total additional $O_2$ production. Again, as in the carbon cycle, the increase in both reduced and oxidised fluxes to the sediments leads to little change in the fraction of S that is buried as pyrite ($f_{py}$), and thus little change to $\delta^{34}S$. This is generally consistent with the geologic record, which shows little change in $\delta^{34}S$ over the Ediacaran[55], especially when considered against the variability window during the Phanerozoic ($\sim$30‰). The record does indicate a shift to slightly higher values towards the Cambrian, which we do not replicate. Nevertheless, a rise in $\delta^{34}S$ would usually be interpreted as an increase in pyrite burial rates relative to gypsum (or reduction in pyrite weathering), thus it is unlikely that the process driving this change could deplete $O_2$. In addition to this, we note that the modelled $f_{py}$ is lower than indicated in some reconstructions[55]. The low $f_{py}$ value is an inherent property of

the COPSE model, which assumes relatively high rates of gypsum weathering and burial. The absolute value of $f_{py}$ is highly uncertain[56], and the important prediction for this study is the change to $f_{py}$ that might be driven by our proposed mechanism, in which our model is in agreement with available data.

## Discussion

We can compare our model atmospheric $pO_2$ predictions to estimates of the oxygen requirements of early animal life forms. This assumes they lived in waters equilibrated with the atmosphere—e.g., in well-mixed (shallow) shelf seas or in benthic slope settings on down-welling margins or high-latitude regions of deep convection[47]. Other locations below the surface mixed layer of the ocean—e.g., seasonally-stratified shelf seas or the open ocean thermocline—tend to be depleted in oxygen due to net respiration of organic matter[47]. It has classically been argued that a minimum oxygen threshold exists for the evolution of animal life[1], but oxygen requirement clearly depends on the type of animal, including their size, mobility, nervous system (information processing) and ecological habits. Sponges (Porifera)—which are the basal animals[57]—have low $pO_2$ requirements[58] $\sim$0.005–0.04 PAL. Hence their evolution was not limited by any of our predicted $pO_2$ levels, consistent with biomarker evidence that demosponges were present by $\sim$660–640 Ma in the Cryogenian[59,60].

The soft-bodied Ediacaran biota $\sim$575–540 Ma are currently interpreted as including a mix of stem- and crown-group animals[61]. The estimated $pO_2$ requirement of soft-bodied, thin, sheet-like animal forms (e.g., *Dickinsonia*), which are assumed limited by $O_2$ diffusion range from $\sim$0.01–0.03 PAL[62] to $\sim$0.06 PAL[63]. Modern benthic invertebrate analogues suggest a requirement of $\sim$0.1 PAL[64], closer to the levels we predict during the early Ediacaran period. Whilst it is tempting to infer that our predicted steady oxygenation during the Ediacaran period enabled the evolution of progressively more complex Ediacaran animals, $pO_2$ may already have been sufficient beforehand. Furthermore, once bilaterian animals with a circulatory system evolved they could have tolerated lower $pO_2$; small bilaterian worms with a circulatory system are estimated to only need $\sim$0.0014–0.0036 PAL[65].

A better case can be made that our predicted rise in $pO_2$ enabled the increased size, mobility, nervous system, carnivory[3,4] and eyesight of animals during the Cambrian explosion—all traits that increase physiological $O_2$ demand. Recent estimates of the minimum oxygen requirement for the evolution and diversification of the Cambrian fauna are $\sim$0.1–0.2 PAL[4] or $\sim$0.25 PAL[5]. Moreover the sequence of appearance of Cambrian fossil animal

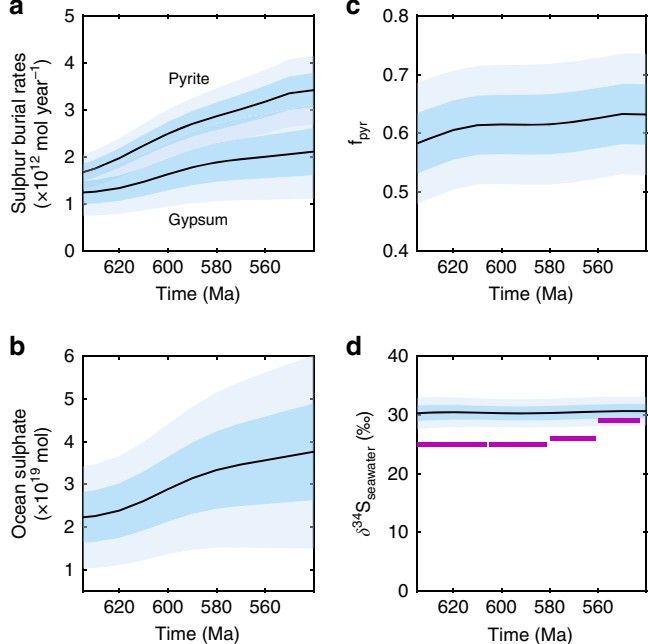

**Fig. 7** Sulphur cycle outputs. **a** Sulphur burial rates. **b** Ocean sulphate concentration, **c** $f_{py}$, **d** $\delta^{34}S$ of seawater. Light shaded windows show ± 1 std. dev, dark shaded windows show ± 0.5 std. dev. Geologic data in purple (averages of [55])

clades transitions from lower to higher oxygen requirement (and decreasing anoxia tolerance) of their extant relatives[5]. Our model analysis predicts that an $O_2$ level below 0.25 PAL is most likely for the beginning of the Ediacaran, and that an $O_2$ level above 0.25 PAL, and perhaps above 0.3 PAL was reached by the end of the period, temporally consistent with the timing of the Cambrian explosion.

Precise quantitative estimates of oxygen levels are difficult to make from a simple box model, in which many parameters are uncertain and one cannot be entirely sure that every relevant process is included. COPSE minimizes this uncertainty as much as possible by comparing its Phanerozoic predictions to multiple whole-Phanerozoic geochemical records[43]. Even so, whilst there remains high confidence in our qualitative result, the quantitative evolution of oxygen levels from ~0.2 to ~0.3 PAL that we show should be taken as an informed best guess.

A simple, unidirectional marine oxygen rise is overly simplistic, and shorter-term oscillations in the redox state of the ocean, at both local and global scales, clearly occurred during the Proterozoic-Phanerozoic transition[2,8,25,66]. Nevertheless, their increasing frequency[16,20] is consistent with an overall trend of oxygenation due to rising atmospheric $pO_2$. To summarise, our predicted long-term Ediacaran oxygenation event driven by increased tectonic degassing is robust to model uncertainties, fits overall trends in geochemical data, and is consistent with existing inferences of $pO_2$ requirements of the Cambrian fauna. Whilst disentangling the many factors influencing faunal evolution is beyond the realms of this study, we provide the first quantitative prediction of Ediacaran oxygenation that is consistent with geo-chemical data and with estimated $pO_2$ requirements for the Cambrian explosion[1,2,5].

## Methods

**Regression analysis of carbonate carbon isotopes.** Supplementary Fig. 1 shows plots of carbonate $\delta^{13}C$ for the Neoproterozoic (left, yellow) and the Ediacaran

subset (right, red)[67]. For both, least-squares regression is performed on the whole dataset and on data that is binned into 5 Myr and 10 Myr groups. The Neoproterozoic data show an average of 2–3‰ and an overall trend towards lower values over time. The Ediacaran data show an average of around 0‰ and no clear trend. Thus we argue that there is no step-change towards more positive values over either the Neoproterozoic Era or the Ediacaran period. This absence of a step-increase is problematic for many proposed mechanisms that infer an increased burial rate of organic carbon relative to carbonates over these intervals (see main text).

**COPSE model alterations.** For this work we use the COPSE (Carbon Oxygen Phosphorus Sulphur Evolution) Reloaded model[43], which is an Earth system box model that computes changes in the C, O, P, S and N cycles over geological timescales to reconstruct long-term climate and sediment geochemistry. It was designed for the Phanerozoic Eon but has been extended previously in simplified forms to test hypotheses for Proterozoic oxygen controls[12,29], reasoning that the same key processes control the major geochemical cycles. Organic carbon burial in the model is dependent on the concentrations of the limiting nutrients nitrogen and phosphorus, with P being the ultimate limiting nutrient. P input is controlled by weathering fluxes, which are dependent on global temperature and also supply alkalinity and cations, resulting in carbonate precipitation. COPSE is a fully-dynamic predictive model in which all fluxes are controlled by the internal pro-cesses, rather than being prescribed. Only external forcings (e.g., degassing rate, uplift rate) are prescribed and the resulting predicted changes in global biogeo-chemical cycling can be tested against isotopic records of carbon, sulphur and strontium[43].

Tectonic changes and biosphere evolution are imposed as external forcings. See Lenton et al.[43] for the Phanerozoic COPSE model runs and comparisons to $\delta^{13}C$, $\delta^{34}S$ and $^{87}Sr/^{86}Sr$ data. In this work, we run the model for the Ediacaran period to test the global response to an increase in $CO_2$ degassing rates. The degassing rate ($D$) forcing is prescribed following a quantitative reconstruction for the late Neoproterozoic[37], and the uplift/erosion ($U$) forcing is assumed to linearly increase based upon the model fit to the strontium isotope system. Land area forcings for weathering of carbonates, granites and basalts are held at their present-day value given the lack of data, and terrestrial biosphere forcings are turned off (as in the Cambrian part of the Phanerozoic run). We run the model 10,000 times using the Monte-Carlo approach of Royer et al.[51]. Full details of the Monte-Carlo parameter changes and procedure are in the main text. Aside from the standard run denoted above, we also run the model with constant uplift, the results of which are shown in the Supplementary Information (Supplementary Note 1, Supplementary Figs. 2–4).

The model equations used in this version of the model are documented below. Whilst a number of external forcings are set for this work, all model processes remain the same as the published model[43], with the exception of the addition of direct mantle input of carbon and reducing power (modelled as $H_2$ gas). We do not include equations relating to the impact of vegetation on the long-term Carbon cycle (i.e., plant-assisted weathering) as these processes were not occurring in the Precambrian and the forcing set used here reduces them to zero. Full model equations including those omitted are documented in Lenton et al.[43], and we direct the reader to this publication for more detail on the model itself. $RO_2$ and $RCO_2$ denote concentrations of oxygen and carbon dioxide relative to present-day. Subscript zeros represent the present-day size of fluxes and reservoirs. Model runs were performed using the MATLAB ODE suite variable timestep solvers[68] in parallel on 6 cores. The COPSE code has been made freely available at https://github.com/sjdaines/COPSE/releases

**List of fluxes**

Phosphorus weathering:

$$phosw = k_{phosw}\left(0.8\left(\frac{silw}{silw_0}\right) + 0.14\left(\frac{carbw}{carbw_0}\right) + 0.06\left(\frac{oxidw}{oxidw_0}\right)\right) \quad (1)$$

P delivery to land surface:

$$pland = k_{11} \cdot V \cdot phosw \cdot \left(k_{aq} \cdot U + \left(1 - k_{aq}\right) \cdot b_{coal}\right) \quad (2)$$

Land organic carbon burial:

$$locb = CP_{land} \cdot pland = k_5 \cdot CP_{land} \cdot pland' \quad (3)$$

P delivery to oceans:

$$psea = phosw - pland \quad (4)$$

Marine new production:

$$newp = r_{C:P} \cdot min(30.9(N/N_0)/r_{N:P}, 2.2(P/P_0)) \quad (5)$$

Marine organic carbon burial:

$$mocb = k_2 \cdot U \cdot \left(\frac{newp}{newp_0}\right)^2 \cdot f(O_2) \quad (6)$$

Optional oxygen dependence for mocb:

$$f(O_2) = 2.1276 \cdot e^{-0.755 \cdot \left(\frac{O_2}{O_{20}}\right)} \quad (7)$$

Marine organic phosphorus burial:

$$mopb = \frac{mocb}{CPsea} \tag{8}$$

Calcium-bound phosphorus burial:

$$capb = k_7 \cdot newp'^2 \cdot f(anox) \tag{9}$$

Iron-sorbed phosphorus burial:

$$fepb = \frac{k_6}{k_1} \cdot (1 - anox) \cdot \left(\frac{P}{P_0}\right) \tag{10}$$

Nitrogen fixation:

$$nfix = k_3 \left(\frac{P - N/r_{N:P}}{P_0 - N_0/r_{N:P}}\right)^2 \text{ for } \frac{N}{r_{N:P}} < P, \text{ else } 0 \tag{11}$$

Marine organic nitrogen burial:

$$monb = \frac{mocb}{CN_{sea}} \tag{12}$$

Denitrification:

$$denit = k_4 \left(1 + \frac{anox}{1 - k_1}\right) \cdot \left(\frac{N}{N_0}\right) \tag{13}$$

Granite weathering:

$$granw = k_{granw} \cdot U \cdot PG \cdot a_{gran} \cdot f_{Tgran} \cdot f_{runoff} \cdot f_{biota} \tag{14}$$

Basalt weathering:

$$basw = k_{basw} \cdot PG \cdot a_{bas} \cdot f_{Tbas} \cdot f_{runoff} \cdot f_{biota} \tag{5}$$

Silicate weathering:

$$silw = granw + basw \tag{16}$$

Carbonate weathering:

$$carbw = k_{14} \cdot C \cdot U \cdot PG \cdot g_{runoff} \cdot f_{biota} \tag{17}$$

Oxidative weathering

$$oxidw = k_{17} \cdot U \cdot g \cdot o^{0.5} \tag{18}$$

Marine carbonate carbon burial:

$$mccb = silw + carbw + mpsb - pyrw - pyrdeg \tag{19}$$

Seafloor weathering:

$$sfw = k_{sfw} \cdot D \cdot e^{k_T^{sfw} \cdot \Delta T} \tag{20}$$

Pyrite sulphur weathering:

$$pyrw = k_{21} \cdot U \cdot pyr \tag{21}$$

Gypsum sulphur weathering:

$$gypw = k_{22} \cdot gyp \cdot U \cdot PG \cdot g_{runoff} \cdot f_{biota} \tag{22}$$

Pyrite sulphur burial:

$$mpsb = k_{mpsb} \cdot \frac{s}{o} \cdot mocb' \tag{23}$$

Gypsum sulphur burial

$$mgsb = k_{mgsb} \cdot s \cdot c_{cal} \tag{24}$$

Organic carbon degassing:

$$ocdeg = k_{13} \cdot D \cdot g \tag{25}$$

Carbonate carbon degassing:

$$ccdeg = k_{12} \cdot D \cdot B \cdot c \tag{26}$$

Pyrite sulphur degassing:

$$pyrdeg = k_{pyrdeg} \cdot D \cdot pyr \tag{27}$$

Gypsum sulphur degassing:

$$gypdeg = k_{gypdeg} \cdot D \cdot gyp \tag{28}$$

**Other calculations**
Atmospheric $CO_2$:

$$CO_2 = a^2 \tag{29}$$

COPSE Reloaded uses the global temperature function from Berner and Kothavala[69] rather than that of Caldeira and Kasting[70].
Global Temperature:

$$\Delta T = k_c \cdot \ln CO_2 - k_l \cdot t/570 \tag{30}$$

where $k_c = 4.328°C$ and $k_l = 7.4°C$

Ocean anoxic fraction:

$$anox = \frac{1}{1 + e^{-k_{anox}\left(k_u\left(\frac{newp}{newp_0}\right) - \left(\frac{O_2}{O_{20}}\right)\right)}} \tag{31}$$

Reduced Gas Flux:

$$rgf = k_{rgf} \cdot D \tag{32}$$

A flux of reduced gas, modelled as $H_2$, is added to the model to explore the possibility that increased subduction and degassing rates may act to lower $O_2$ levels by delivering more reductant from the mantle. The rate of input is defined by an assumed present-day rate ($k_{rgf}$) and scales with the relative degassing rate. To maintain balance of the oxygen cycle at present day, the present day oxidative weathering flux is reduced by $k_{rgf}$. This modification, in turn, requires an additional source of carbon to the surface system to maintain balance, which is represented by mid-ocean ridge degassing of $CO_2$ with magnitude equal to $k_{rgf}$. We also assume that the large sedimentary reservoirs in the model (C, G, PYR, GYP) do not change over time. This avoids further model extension to link these reservoirs to the mantle[71] and is justified because the changes in the relative sizes of these reservoirs over the timescales we wish to consider are negligible (an average of 2.5% change over 100 Myrs in the model of Hayes and Waldbauer[71]).

**Strontium isotope system**. This study follows Lenton et al.[43] whereby the strontium cycle and its isotopes are implemented following Francois and Walker[72] and Vollstaedt et al.[73] with some improvements to the formulation described in Mills et al.[74].
**Strontium fluxes**
Mantle Sr Input:

$$Sr_{mantle} = k_{Srmantle} \cdot D \tag{33}$$

Basalt weathering input:

$$Sr_{basw} = k_{Srbasw} \cdot \frac{basw}{k_{basw}} \tag{34}$$

Granite weathering input:

$$Sr_{granw} = k_{Srgranw} \cdot \frac{granw}{k_{granw}} \tag{35}$$

Inputs from carbonate sediments

$$Sr_{sedw} = k_{Srsedw} \cdot \frac{carbw}{k_{carbw}} \cdot \frac{SSr}{SSr_0} \tag{36}$$

Burial in carbonate sediments:

$$Sr_{sedb} = k_{Srsedb} \cdot \frac{mccb}{k_{mccb}} \cdot \frac{OSr}{OSr_0} \tag{37}$$

Removal in seafloor weathering:

$$Sr_{sfw} = k_{Srsfw} \cdot \frac{sfw}{k_{sfw}} \cdot \frac{OSr}{OSr_0} \tag{38}$$

The relative proportions of the burial and seafloor weathering removal fluxes of strontium are assumed to follow the same proportions as the corresponding fluxes in the carbon system, with the total flux dictated by assuming present-day steady state for oceanic Sr concentration.
Output from metamorphism:

$$Sr_{metam} = k_{Srmetam} \cdot D \cdot \frac{SSr}{SSr_0} \tag{39}$$

Although there is no fractionation of Sr isotopes associated with the input and output fluxes to the ocean, decay of $^{87}Rb$ to $^{87}Sr$ influences the $^{87}Sr/^{86}Sr$ ratio over long timescales (and is responsible for the differing $^{87}Sr/^{86}Sr$ values between different rock types). The decay process is represented explicitly in the model:

$$^{87}Sr/^{86}Sr_{granite} = {}^{87}Sr/^{86}Sr_0 + {}^{87}Rb/^{86}Sr_{granite}\left(1 - e^{-\lambda t}\right) \tag{40}$$

$$^{87}Sr/^{86}Sr_{basalt} = {}^{87}Sr/^{86}Sr_0 + {}^{87}Rb/^{86}Sr_{basalt}\left(1 - e^{-\lambda t}\right) \tag{41}$$

$$^{87}Sr/^{86}Sr_{mantle} = {}^{87}Sr/^{86}Sr_0 + {}^{87}Rb/^{86}Sr_{mantle}\left(1 - e^{-\lambda t}\right) \tag{42}$$

Where time ($t$) is in years from Earth formation (taken to be 4.5 billion years ago). For each rock type, the rubidium-strontium ratio is then calculated such that the observed present-day $^{87}Sr/^{86}Sr$ ratio is achieved for each rock type after 4.5 billion years:

$$^{87}Rb/^{86}Sr = \frac{\left(^{87}Sr/^{86}Sr_{present} - {}^{87}Sr/^{86}Sr_0\right)}{\left(1 - e^{-\lambda \cdot 4.5 \times 10^9}\right)} \tag{43}$$

The isotopic composition of the ocean and the sediments are calculated by first creating reservoirs consisting of Sr concentrations multiplied by their isotopic ratios, where $\delta Sr_X$ denotes the $^{87}Sr/^{86}Sr$ ratio of reservoir $X$:

$$\frac{d(OSr \cdot \delta Sr_{ocean})}{dt} = Sr_{granw} \cdot \delta Sr_{granite} + Sr_{basw} \cdot \delta Sr_{basalt}$$
$$+ Sr_{sedw} \cdot \delta Sr_{sediment} + Sr_{mantle} \cdot \delta Sr_{mantle} - Sr_{sedb} \cdot \delta Sr_{ocean}$$
$$- Sr_{sfw} \cdot \delta Sr_{ocean} \tag{44}$$

$$\frac{d(SSr \cdot \delta Sr_{sediment})}{dt} = Sr_{sedb} \cdot \delta Sr_{ocean} - Sr_{sedw} \cdot \delta Sr_{sediment} -$$
$$Sr_{metam} \cdot \delta Sr_{sediment} \tag{45}$$

The ocean $^{87}Sr/^{86}Sr$ ratio is calculated by dividing the new reservoir by the known concentration:

$$\delta Sr_{ocean} = \frac{OSr \cdot \delta Sr_{ocean}}{OSr} \tag{46}$$

The carbonate sediment $^{87}Sr/^{86}Sr$ ratio includes an additional term to account for rubidium decay within the sedimentary reservoir:

$$\delta Sr_{sediment} = \frac{SSr \cdot \delta Sr_{sediment}}{SSr}$$
$$+ {^{87}Rb}/{^{86}Sr_{carbonate}} \left(1 - e^{-\lambda \Delta t}\right) \tag{47}$$

Where here $\Delta t$ is time elapsed since the start of the model run. The rubidium-strontium ratio of sediments is calculated to achieve the average crustal $^{87}Sr/^{86}Sr$ of 0.73[75].

## Data availability
The model data that support the findings of this study are available from the corresponding author upon reasonable request.

## Code availability
The COPSE code is freely available at https://github.com/sjdaines/COPSE/releases

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

## Acknowledgements

JJW was funded by a UK Natural Environment Research Council (NERC) doctoral training partnership grant (NE/L002558/1). BJWM is funded by NERC (NE/R010129/1) and by a University of Leeds Academic Fellowship. TML was supported by NERC (NE/P013651/1).

## Author contributions

J.J.W., B.J.W.M. and T.M.L. contributed to the design of the study and writing of the paper. J.J.W. performed model simulations.

## Additional information

**Competing interests:** The authors declare no competing interests.

