## [Peer Review File · Nature Communications]

Reviewers' comments:

Reviewer #1 (Remarks to the Author):

Williams and colleagues present a biogeochemical modeling study in which it is argued that a secular increase in rates of volcanic CO₂ degassing during the late Neoproterozoic and Paleozoic stimulated an increase in organic carbon burial fluxes and a net accumulation of O₂ in the atmosphere, with atmospheric pO₂ estimated to have approximately doubled during the Ediacaran. These quantitative estimates are then linked to the presumed requirements of early metazoan organisms and the emergence of later mobile predators during the Cambrian explosion.

Although I am sympathetic to the authors' aims here, and consider the proposed link between CO₂ degassing and atmospheric O₂ interesting (if somewhat self-fulfilling given the design of the model), I have a few concerns about how the model is presented and how the results are discussed that should preclude publication of the manuscript in its current form.

One of my primary concerns involves analyzing and explicitly evaluating uncertainty in the model results. The model consists of 14 prognostic equations, along with 39 parameterizations, some of which are mechanistically well-established but many of which are much less certain. All models are idealized and uncertain to some extent, and in studies like this one should be permitted to rely on parsimony when possible. However, the most important main text figure (Fig. 4) shows the results from only three 'single scenario' analyses with absolutely no uncertainty reported. Figure 1 shows an error bound on the proxy used for degassing rates (and Fig. S5 shows attendant uncertainty in rates of degassing), but this error bound is not justified mechanistically (actually, it is explicitly acknowledged as 'arbitrary'; SI Lines 266-267). This is important, as a perfectly plausible trajectory through the uncertainty range in subduction zone length shown in Fig. 1 is no change at all, and this needs to be explicitly ruled out if the ensuing narrative is to have any validity. There is some discussion of Monte Carlo analysis with regard to tectonic forcing in the supplement, but only 100 runs are performed and there is no discussion as to whether this should be sufficient to capture the underlying uncertainty in the model, nor are the results of this Monte Carlo analysis presented explicitly in the main text or the supplement as far as I can tell.

Second, all of the model scenarios outlined require a large gypsum burial flux. Under the assumption of constant uplift and increasing degassing (Fig. S7), the relative pyrite burial fluxes (f_{py}) range between ~ 0.4 - 0.5 , while under the assumption of constant outgassing and dynamic uplift (Fig. S12) f_{py} values range between ~ 0.4 - 0.6 . Yet, somewhat strangely, this result is never discussed in the context of the sulphur isotope record. Most reconstructions for this period indicate that pyrite burial totally dominates removal fluxes of sulphur from the ocean, which at face value is a serious red flag

for the model results presented here. Such isotopic records are not perfect, of course, but it is intriguing that the authors interpret the carbon isotope record at face value (which is also not without its problems) while completely ignoring the sulphur isotope record. The authors really need to discuss the sulphur isotope record explicitly, why they see deviations from the conventional view in their model results, why they don't consider this an issue for their interpretations, and finally why they consider the carbon isotope record to have significantly more fidelity than the sulphur isotope record. I consider it plausible that all of these issues can be evaluated mechanistically, but it is unacceptable for them to be ignored entirely. Indeed, it is particularly striking that the authors present selenium isotope data, a system that most geochemists would agree is relatively poorly understood at present, but no discussion of contemporaneous sulphur isotope data.

Third, the way that geochemical proxies are dealt with in Fig. 1 of the main text is cursory to the point of being almost meaningless. What does "Earth system oxygenation" mean? This is fine for the first sentence or two of a caption, but if not elaborated on is not very useful. Selenium isotopes, cerium anomalies, and iron speciation all respond to completely different geochemical conditions, are preserved in different sedimentological archives, and exist along varying degrees in the trajectory toward being well-established geochemical proxies. Even within a single proxy system the figure is frustratingly vague - what does "deep water O₂" mean, for instance? To what extent does absence of evidence amount to evidence of absence in this regard? This point is particularly important given the implication of the figure that deep water O₂ only emerges after the secular increase in degassing hypothesized by the authors. How well-constrained is this?

Lastly, I can't help but wonder if the manuscript would be more effectively argued and impactful in a longer format (even if only considerably lengthened as a Nature Communications submission, for which I think there is quite a bit of scope given the current brevity?). The model, despite being low-order, is complex, and even though there is little uncertainty analysis in the current manuscript there are already 16 supplementary figures. The basic model framework has been presented elsewhere (Bergman et al., 2004; Lenton et al., 2018), and I appreciate that there are editorial constraints on the short form. But I would argue that the paper would be much more convincing and impactful beyond the relatively small modeling community if more of this were to be made transparent in the main text of the paper. This is particularly true given that the basic result - that increasing CO₂ degassing leads to an increase in atmospheric O₂ - is a rather obvious result based on the way the model is constructed.

Minor comments:

- It would be helpful if the panels in Fig. 3 were labeled in the figure itself to correspond with the caption text.

- Panels in Fig. 4 should also be labeled. Also, the discussion around Lines 141-145 does not square with the panels shown in the figure itself. Double-check this.

- References 45 and 50 are the same paper (though one is in the bibliography as "in press")

Reviewer #2 (Remarks to the Author):

This manuscript by Williams et al. presents a new application to the COPSE base model to test whether a change in plate tectonics (i.e. subduction rates) would have an effect on atmospheric oxygen levels. Building on much of this workgroup's prior work, they show that oxygen levels increased throughout the late Neoproterozoic, reaching levels that are thought to be high enough ($pO_2 = \sim 0.1$ PAL) to support the animal life that comprise the Ediacaran fauna. The driver for this oxygenation event relies upon the assumption that increased weathering rates, driven by an increase in degassing of CO_2 from higher subduction rates, delivered more phosphate to the oceans, which elevated primary production and organic burial rates (i.e. a net increase in atmospheric O_2). Though this seems like a long line of assumptions to include in a model, the authors at least acknowledge that parameter constraints during this time are not well known. These model predictions are consistent with other geochemical studies that suggest oxygen levels were increasing during this time, as well as being above the minimum levels needed to sustain metazoan life. Thus, I'm not too surprised when the curve for subduction length (Fig. 1) looks almost identical to the O_2 curve (Fig. 4).

This is a fairly well written manuscript, though it does have some editorial errors (see below) and the modeling appears to be carefully done with appropriate sensitivity testing, but I have some concerns that cause me to recommend that this manuscript not be accepted in its current form. I'm not an expert in the COPSE modeling approach, so I'll defer to the other reviewer's expertise in critically reviewing this model and the inherent assumptions. However, a read through the text and supplement have raised a few concerns that I think the authors should address and state more clearly in the main text.

First, I'm a bit surprised that the authors seem to discount the strong control that the sulfur cycle plays on atmospheric O_2 levels via pyrite burial. This version of the model appears to work on the basis that pyrite burial is reciprocal to that of organic burial (I'm probably overstating this, you say balanced by in the text) to regulate atmospheric O_2 levels, a process that the authors take from a paper by Bob Berner written 35 years ago (Berner and Raiswell, 1983). However, it appears that in some of Berner's later work (and many others) that the sulfur cycle plays a much more important role in affecting atmospheric O_2 than just the carbon cycle. For example, in the first version of the GEOCARBSULF model (Berner, 2006), 1 mol of O_2 is produced for every mol of organic carbon that is buried, but nearly 4 mols of O_2 are produced with one mol of pyrite that is buried (ignoring the organic carbon used during microbial sulfate reduction). Is this wrong? If so, correct this line of

reasoning in the text. In the supplement the authors present their model estimates of the sulfate reservoir size and pyrite burial rates, but these results all seem to be based on the organic burial rate (which is derived from oceanic phosphate concentrations, which are controlled by silicate weathering rates, which are controlled by CO₂ levels, which are controlled by degassing as a function of subduction rates, which are controlled by mid-ocean ridge length). As I stated above I am no expert on the COPSE model, so for folks like me I would like to see the authors explain more clearly why the sulfur cycle isn't as important as they think, or explain why my example above is incorrect. Maybe it's a matter of the sulfate reservoir is much smaller than today?, but I would think that it should still play an important role. It would be helpful to see some model runs where this 'canceling out' of pyrite burial and oxygen release wasn't parameterized in this way (Equation 33). What if pyrite burial contributed XX amount of atmospheric O₂?

Secondly, and maybe this is my problem and not keeping up with all of the latest papers from this prolific workgroup and understanding the importance of the COPSE model, but after a few reads of this manuscript and I find myself left thinking that this approach uses more assumptions and guesses at parameter values than known values. I understand that it is impossible to know what uplift rates were like 600 million years ago.... Or today for that matter, but I'm left thinking that the modeled O₂ curve is simply a result of some calculations using constants values/rates on the subduction curve in Figure 1. Not to diminish the importance or impact of this modeling work, which I appreciate is difficult to do and assign values to forcing factors, but the modeled O₂ curve is essentially the same shape as the subduction curve with a slight bend during the Ediacaran to account for increased uplift rates associated with the formation of Pannotia. This basically suggests that tectonics controls everything, which I suppose it does, but I suspect there is more to the story than just subduction zone length. I'm not sure I have a good answer for how to placate this critique other than to say I would like to see more effort put into constraining the model at certain time points. For example, the inflection in uplift appears to be made based on sediment abundance at the end Proterozoic, but are there other estimates the authors can draw upon? Perhaps the database hosted on Macrostrat.org (age and rock type) can provide some constraints, or at least be used in sensitivity testing? Surely this would affect the volume of rock being weathered... which should have different average phosphate concentrations. Or, perhaps the rates can be adjusted slightly at different time points to fit the ⁸⁷Sr/⁸⁶Sr curve. My relatively uninformed read of the model description is that a few rates/constraints were chosen from previous work, and because the model estimates kind of match geochemical data measured in other studies (e.g. Fig. 3), it's close enough. Perhaps if this 'ground truthing' the model isn't possible, then I feel that the authors should more clearly state the inherent assumptions in this approach. It wasn't clear to me in my first read that this model assumes a lot, including: a constant supply of phosphate, constant weathering rates, a constant mix of rock with constant phosphate contents, phosphate is the only nutrient needed to produce a constant amount of primary productivity, and constant organic burial rates uniformly in the oceans. I have no evidence to say whether this is wrong or inaccurate, but I think it'd be better to at least let the reader know how many assumptions are made with this approach.

That said I found some other minor issues with the manuscript that require some attention spelled out below. I find that this manuscript has potential to be an important contribution to the field, particularly if the authors can underscore the major assumptions that the COPSE model is based on and that it isn't really constrained by highly resolved measurements.

- Line 9: Please use the ages from most recent Geologic Timescale, it's 2018 after all. The end of the Cambrian is 485.4 Ma. Also check that the ages for the Neoproterozoic periods are those in the 2012 GTS.

- This may be a style thing, but when discussing time, one should use the word 'during' instead of 'in'. For example, the first sentence of the abstract should end as "~850 to 541 Ma during the Neoproterozoic Era." This issue is present throughout the manuscript.

- Figure 1: This figure is incomprehensible without some more explanation in the text, figure caption, and/or figure itself. It's a bit odd that Fig. 1B is explained in the main text before Fig. 1A, but what is more puzzling is what the Se and Ce data indicate. Which end of this figure is more oxic or anoxic? What does the dashed blue line mean? What does the "Deep water O₂" mean, from who's study, and why is this plotted on the figure showing subduction zone length? What does the gray area mean around the black line in 1A? A little annotation to the figure and caption will go a long way to help the reader understand what is to be communicated here.

- Line 53: Please explain the last part of this sentence. Why does organic burial only temporarily increase atmospheric O₂? I cannot follow.

- Line 58: A transition is needed here, or perhaps introduce this notion earlier in this paragraph.

- Line 63: An alternative what?

- Line 64: Can you include these data in Fig. 1A? It seems like this would strengthen your argument that geologic data and modeling results are consistent with each other, which makes it seem reasonable that subduction zone length can be used as a proxy in other ways.

- Lines 78–79: Late and early should be lower case. These are not formal time intervals (unlike the Early Ordovician, for example). Please make the appropriate changes throughout the rest of the text.

- Line 86: Seems like 'protected' is an odd word choice.

- Lines 94–98: This assumes fractionation is constant, that there isn't a fractionation effect associated with carbonate formation, and that there is complete oxidation of organic matter.

- Line 108: What effect?

- Line 109: Might be a good place to remind the reader what reduced source you're referring to, especially if sulfur isn't as important in this model.

- Line 112: What does 'sped up' mean? Doubling? Faster burial and weathering rates?

- Line 115: Does this mean that atmospheric CO₂ levels were constant? I guess not as this figure is buried in your supplement. I would suggest you promote this figure (and a sulfur reservoir/pyrite burial plot) to the main text. Do these values correspond to other CO₂ estimates? How do they agree with the timing of the glaciations during this interval? I thought that the way we got out of these glaciations was because CO₂ levels increased greatly during this time, but you show there's a steady decrease. Something is off here and you should probably acknowledge this discrepancy.

- Line 142: Are these the correct figure references?

- Line 143: Missing superscripts here.

- Lines 158–160: Wait.... How can pyrite burial be suppressed by rising O₂ levels (re-oxidation?) but organic burial can occur unchecked? It seems to me that burial means burial and that either or both can occur without later re-oxidation. As stated above, doesn't pyrite burial yield 4 times as much O₂ compared to organic burial? It seems like this might be a good place to differentiate the two models and why GEOCARBSULF and COPSE are so different with respect to how they deal with the sulfur cycle.

- Figure 4: Again, this figure needs better annotation. There aren't letters associated with the different panels. Why show the data from McArthur et al. but refer to the data from Cox et al. in the main text? Why not both? Superscripts missing for ⁸⁷Sr/⁸⁶Sr and the wrong symbol is used in 'δ¹³Ccarb' in the caption.

- Lines 210–213: This is a fairly subjective claim to make that measured $\delta^{13}\text{C}$ values don't change throughout this interval. $\delta^{13}\text{C}$ values are all over the place and rarely at 0‰. One could argue that there are several baselines that are achieved (+4‰ from 700–650 Ma, -1‰ from 650–610 Ma, +4‰ from 610–580 Ma, etc.). Could these shifting baselines reflect short-term changes in weathering, degassing, or organic burial rates? Do you actually calculate the average value over this time and find that it is 0‰ to agree with your model? I'd be surprised if the average value isn't more like +2‰. What would it mean in your model (i.e. sensitivity testing) if the average $\delta^{13}\text{C}$ value was -2 or +2‰? How might that affect organic burial rates and ultimately O₂ levels? Ultimately I think it's too much of a simplification to say that the long-term $\delta^{13}\text{C}$ average is 0‰, particularly when these excursions are thought by many to reflect major changes to the redox state of the oceans.

- Line 231: EdiacaraN?

Reviewer #3 (Remarks to the Author):

Review of "A Tectonically Driven Ediacaran Oxygenation Event" by Williams, Mills, and Lenton

In this study, the COPSE model is used to test a new hypothesis for the Neoproterozoic Oxygenation Event (NOE). The paper argues that a ~1.7 fold increase in crustal production between 720 and 540 Ma drove an increase in carbon dioxide outgassing, and that this necessitated greater total carbon burial to balance the carbon cycle. Although the proportion of carbon buried as organic carbon did not necessarily change, elevated total burial resulted in greater organic burial, and it is the absolute organic burial flux that determines the oxygen source flux. The COPSE model outputs suggest that this increase in the oxygen flux was sufficient to approximately double the atmospheric O₂ content, which could have enabled the appearance and diversification of complex animals in the Cambrian. It is also argued that their COPSE model outputs track the observed Sr isotope record, although it is necessary to impose a sizeable change in erosion/uplift to achieve this.

This is an interesting study that presents a novel hypothesis to explain the NOE. The paper is commendable for testing the self-consistency of their idea with a complete biogeochemical model rather than merely presenting back-of-the-envelope calculations that can overlook important feedbacks and constraints. The paper is generally easy to follow and, with a few minor exceptions that could be easily remedied (see below), the model is described in sufficient detail for other researchers to try and reproduce the general findings.

However, I cannot recommend publication at present for several reasons enumerated as follows.

1) The main issue I have with this paper is that there are a large number of parameters, parameterizations, and coefficients that are not well constrained so that the conclusions may not be accurate and robust. The paper claims (line 255) that “our predicted long-term Ediacaran oxygenation event driven by increased tectonic degassing is robust to model uncertainties”. This has not been demonstrated in the paper. There should be some sort of sensitivity test with key unknown parameters - ideally some kind of Monte Carlo analysis that runs the model thousands of times, sampling a broad range of plausible parameter values. Alternative parameterization functions should also be considered. In the paper abstract it is claimed that pO₂ doubled during the Ediacaran. The paper needs to show that this quantitative statement holds true for a wide range of model assumptions, and not just for the somewhat arbitrary parameterizations and parameter values adopted in their nominal model. To offer some concrete examples, there are two questionable model assumptions that could, if modified, potentially change the paper’s conclusions about pO₂.

2a) The first questionable assumption relates to the parameterization of seafloor weathering (equations 30a and 30b in the supplementary material). This parameterization implies seafloor weathering is purely temperature dependent, with no dependence on crustal production or ocean pH. The pH dependence is arguably small enough to be neglected, but the crustal production dependence is not. If subduction zone length increased in the Neoproterozoic as the authors claim, then there would presumably be more seafloor basalt available for weathering, and therefore increased uptake of CO₂ as carbonate in the seafloor. Indeed, many previous carbon cycle models have included a linear dependence on crustal production or spreading rate (e.g. Sleep and Zahnle 2001, JGR Res; Franck et al. 2002, Tellus B; Le Hir et al. 2008, Geology; Foley 2015, ApJ) or a power law dependence that encompasses linear proportionality (Krissansen-Totton et al. 2018 PNAS). This is potentially important because if the actual response to elevated degassing is increased carbonate uptake in the seafloor, then the proportion of carbon buried as organic carbon may drop, and the rise in oxygen may be diminished. For example, the paper’s initial (preset day) value for seafloor weathering is 1.75 Tmol C/yr. If this increased by ~1.7x during the Ediacaran due to a linear dependence on crustal production, then the final seafloor weathering flux (before accounting for T-dependence) will be $1.75 \times 1.7 = 3.0$ Tmol C/yr. Thus there is a potential ~1.2 T mol C/yr increase in carbonate burial in the seafloor that could offset the increase in organic burial.

2b) This effect is compounded by the paper’s choice of a relatively weak temperature dependence for seafloor weathering (equation 30a). Coogan and Dosso (2015, EPSL), and Krissansen-Totton and Catling (2017, Nat. Comm.) both suggest that stronger temperature dependencies are possible. This would also have the effect of producing a large seafloor weathering response to an increase in degassing which would act to offset the organic burial increase.

2c) Furthermore, differing seafloor weathering parameterizations will also affect the Sr isotope model. The model fit to the Sr isotope record could potentially be worsened by including a crustal production dependence and a stronger temperature dependence for seafloor weathering. This should be investigated.

3) The second questionable model assumption I noticed was the choice to ignore the increased flux of reduced gases associated with elevated carbon dioxide degassing. For the same mantle redox conditions, higher CO₂ outgassing will also result in a greater flux of reduced gases such as H₂ and CO, which are oxygen sinks and act to offset the pO₂ increase caused by the higher throughput of carbon. The paper acknowledges this possible sink on lines 225-228 of the supplementary materials, but then argues that it is unimportant because the reduced gas flux is much smaller than organic weathering and other oxygen sinks. Canfield (2005, Annual Review EPS) is cited to support this claim, although I could not find specific numbers in this Canfield paper. Canfield (2006, Phil. Trans. Royal Soc. B) estimates the modern H₂ flux as 1.8-5.0e11 mol/yr, which is indeed small compared to the 7.75e12 mol/yr present day oxidative weathering flux assumed by the authors. However, the H₂ flux from Canfield (2006) is at the low end of estimates from the literature. Other estimates for modern H₂ outgassing range from 0.9e12 mol/yr to 2.2e12 mol/yr (see Holland 2009 GCA; Catling and Kasting 2017, Atmospheric Evolution on Inhabited and Lifeless Worlds, p. 208-210). If H₂ outgassing were at the upper end of this range and responded linearly to total degassing then the O₂ sink would increase by $2.2 * 1.7 - 2.2 = 1.5$ Tmol /yr over the Ediacaran. This would also dampen the rise in oxygen somewhat.

4a) Finally, I wonder to what extent the paper's choices of absolute present day fluxes affects the conclusions. Although it is not plotted or stated anywhere, I tentatively infer that the assumed modern day value for carbonate weathering used in this study is 6.25 Tmol/yr (based on steady state values in page 6 in supplement). Estimates in the literature range from 7-14 Tmol/yr (Hartmann et al. 2009, Glob. Planet. Change). The same paper lists a wide range of estimates for silicate weathering. The present manuscript also adopted a modern total outgassing flux of 7.9 Tmol/yr, whereas literature estimates range from 4-10 Tmol/yr (Lee and Lackey 2015, Elements; Berner 2004, The Phanerozoic Carbon Cycle). Similar uncertainties exist for the organic carbon fluxes, although I have not dived into the literature on this. The submitted manuscript should test the sensitivity of the results to changing present day fluxes within plausible bounds.

4b) A related issue is whether it is reasonable to assume carbonate and organic carbon degassing increases linearly with subduction zone length. Intraplate volcanism on the modern Earth is potentially quite large (Lee and Lackey 2015, Elements) and so total degassing may respond less than linearly to elevated rates of subduction. This would potentially lessen the change in carbon throughout and therefore lessen the Ediacaran rise of oxygen.

To be clear, I'm not saying the proposed oxygenation mechanism doesn't work. The paper provides a scenario that might explain both the Neoproterozoic rise of oxygen and several other geochemical data sets. However, it's not enough to merely show such a scenario is possible. The paper needs to demonstrate that these conclusions are robust. Specifically, if a more standard seafloor weathering parameterization is adopted, if elevated H₂ fluxes are accounted for, and if broader ranges of present day fluxes are considered, do the paper's conclusions about oxygen still stand? The paper could be made more convincing and impactful by demonstrating that an approximate doubling of pO₂ occurs despite these uncertainties.

Other important issues:

5) How exactly is pO₂ through time being calculated? As far as I could see there is no differential equation that specifies the time-evolution of the atmospheric oxygen reservoir. The reader has to piece it together from Figure 2 and the various carbon flux equations. Please explain clearly how the atmospheric oxygen reservoir is calculated since this is crucial to the paper's main conclusions. Also, please explain the k_{O₂} term in equation 37 in the supplementary material.

6) Lines 50-61. I think this paragraph is too dismissive of previously proposed mechanisms for the NOE. The paper argues that all these mechanisms imply a rise in d¹³C of carbonates, and that such a rise is not observed in the isotope record. However, Krissansen-Totton et al. (2015, *AJS*) found that there is a statistically significant increase in both d¹³C(carbonates) and steady state fractional organic carbon burial between the mid Proterozoic and Phanerozoic. The magnitude of this fractional organic burial increase is around 30%, albeit with large uncertainties. A 30% increase in fractional organic burial is comparable to the increase in the paper's model (Fig. S15A), which is enough to cause an approximate doubling in pO₂. This increase in steady state d¹³C is not visible in the 550-700 Ma record plotted in this study because the broader context of the time series is omitted. Perhaps the observed increase in d¹³C is insufficient for the alternative NOE hypothesis summarized in lines 50-61, but it is inaccurate to say there has been no change in d¹³C and imply that the other proposed mechanisms that predict d¹³C increases do not work. Other possibilities for decoupling organic burial and the d¹³C record have also been proposed and ought to be mentioned (e.g. Schrag et al. 2013, *Science*).

7) The incorrect claim that the d¹³C record has been constant is repeated on line 112 and again on lines 145. Relatively modest changes in fractional organic burial can cause substantial changes in pO₂, and such changes are consistent with the carbon isotope record. Statements about the constancy of the carbon isotope record should be adjusted accordingly.

8) The descriptions of the degassing forcing (Supporting Information 3) and the uplift forcing (Supporting Information 4) are somewhat confusing. For instance, the paper states that 100 Monte Carlo simulations were run, and the 1-sigma and 2-sigma standard deviations were calculated. However, Fig. S5 shows 20% error bounds, not 1-sigma or 2-sigma errors. If the dashed lines are 20% errors, this seems to suggest the 2-sigma envelope is quite large and potentially consistent with constant relative degassing. Is this interpretation correct? If so, this should be clearly stated in the main text.

9) Similarly for the uplift forcing, the plotted 20% errors suggest that the 2-sigma envelope is very large. I also don't understand how the 750-550 Ma portion of the curve is derived from (or consistent with?) the scarce supercontinent and orogeny constraints in this time span.

10) More generally, what is the purpose of these error calculations anyway? It looks like only the median degassing and uplift curves are being used as input for the COPSE model calculations.

Minor issues/corrections

11) The specified range for Proterozoic oxygen on line 27 is a little narrow (compare Lyons et al. 2014, Nature).

12) Please explain all lines, symbols and shading in the Figure 1 legend. In general, figure legends could be more descriptive and would help improve the clarity of the manuscript.

13) Line 142 references Fig. 4c when discussing fractional organic burial, but Fig. 4c is a plot of Sr isotopes.

14) Line 205: Refer the reader to the supplementary materials that explain where the uplift curve comes from. On first reading it isn't clear why this uplift curve should be believed.

15) Supplementary material: there are many equations for plant assisted weathering, organic burial on land, vegetation feedbacks etc. I assume these equations aren't used for the version of model adopted in this paper since there was no vegetation in the Precambrian. Perhaps they could be omitted so as not to overwhelm and confuse the reader with unnecessary equations (i.e. only include the pre-plant versions of these equations)?

16) Supplementary material lines 14-16: It might be worth citing Coogan and Dosson (2015, EPSL) here to justify the temperature dependence of seafloor weathering.

17) Supplementary equations lines 48 and 49: could you write out these weathering equations to remove any possible ambiguity?

18) Supplementary equations 24, 25, and 27: PG appears to lack a definition (unless missed).

19) Supplementary lines 81-83 simply repeat supplementary lines 73-75.

20) Fig S3 and S4: Please adjust all large-numbered contours to be in exponent form. Nobody likes counting zeros.

21) Why is the organic carbon burial curve in Fig. S15 different to that in Fig. S8? The difference between modeled degassing and dynamic degassing isn't explained clearly anywhere (I searched "dynamic" and it is only mentioned in the supplementary figures). This is another area where more descriptive figure captions would be helpful.

22) Finally, if feasible, the authors should make the version of COPSE used for this study publicly available and provide a link to its location in the manuscript. This would greatly improve the transparency and reproducibility of their work.

Reviewers' comments:

Reviewer #1 (Remarks to the Author):

Williams and colleagues present a biogeochemical modeling study in which it is argued that a secular increase in rates of volcanic CO₂ degassing during the late Neoproterozoic and Paleozoic stimulated an increase in organic carbon burial fluxes and a net accumulation of O₂ in the atmosphere, with atmospheric pO₂ estimated to have approximately doubled during the Ediacaran. These quantitative estimates are then linked to the presumed requirements of early metazoan organisms and the emergence of later mobile predators during the Cambrian explosion.

Although I am sympathetic to the authors' aims here, and consider the proposed link between CO₂ degassing and atmospheric O₂ interesting (if somewhat self-fulfilling given the design of the model), I have a few concerns about how the model is presented and how the results are discussed that should preclude publication of the manuscript in its current form.

We have extensively revised the manuscript to address the reviewer's concerns.

One of my primary concerns involves analyzing and explicitly evaluating uncertainty in the model results. The model consists of 14 prognostic equations, along with 39 parameterizations, some of which are mechanistically well-established but many of which are much less certain. All models are idealized and uncertain to some extent, and in studies like this one should be permitted to rely on parsimony when possible. However, the most important main text figure (Fig. 4) shows the results from only three 'single scenario' analyses with absolutely no uncertainty reported. Figure 1 shows an error bound on the proxy used for degassing rates (and Fig. S5 shows attendant uncertainty in rates of degassing), but this error bound is not justified mechanistically (actually, it is explicitly acknowledged as 'arbitrary'; SI Lines 266-267). This is important, as a perfectly plausible trajectory through the uncertainty range in subduction zone length shown in Fig. 1 is no change at all, and this needs to be explicitly ruled out if the ensuing narrative is to have any validity. There is some discussion of Monte Carlo analysis with regard to tectonic forcing in the supplement, but only 100 runs are performed and there is no discussion as to whether this should be sufficient to capture the underlying uncertainty in the model, nor are the results of this Monte Carlo analysis presented explicitly in the main text or the supplement as far as I can tell.

This is a useful criticism and we have greatly improved the uncertainty analysis in the revision. We now use the most recently published 'COPSE reloaded' model (Lenton et al., 2018) and perform 10,000 runs following the Monte-Carlo setup of Royer et al (2014), who have analysed a similar type of model with many of the same relationships. We have tested the most important parameters that might work against our hypothesis (as laid out by reviewer 3) as well as the range of uncertainty in our degassing forcing. The 'arbitrary' error bound plotted in the SI has been replaced with the error bound given in the paper that presents the degassing rate information we use, as should have been the case all along. On the prospect of a 'no-change in degassing scenario', we have noted that whilst this is possible, it is very unlikely given the need to move from opposite ends of the uncertainty window, and given the multiple independent estimates for a rise in degassing rates at this time (for which we cite the compilation in Mills et al. 2017). This more detailed uncertainty analysis provides further support for our proposal.

Second, all of the model scenarios outlined require a large gypsum burial flux. Under the assumption of constant uplift and increasing degassing (Fig. S7), the relative pyrite burial fluxes (f_{py}) range between ~ 0.4 - 0.5 , while under the assumption of constant outgassing and dynamic uplift (Fig. S12) f_{py} values range between ~ 0.4 - 0.6 . Yet, somewhat strangely, this result is never discussed in the context of the sulphur isotope record. Most reconstructions for this period indicate that pyrite burial totally dominates removal fluxes of sulphur from the ocean, which at face value is a serious red flag for the model results presented here. Such isotopic records are not perfect, of course, but it is intriguing that the authors interpret the carbon isotope record at face value (which is also not without its problems) while completely ignoring the sulphur isotope record. The authors really need to discuss the sulphur isotope record explicitly, why they see deviations from the conventional view in their model results, why they don't consider this an issue for their interpretations, and finally why they consider the carbon isotope record to have significantly more fidelity than the sulphur isotope record. I consider it plausible that all of these issues can be evaluated mechanistically, but it is unacceptable for them to be ignored entirely. Indeed, it is particularly striking that the authors present selenium isotope data, a system that most geochemists would agree is relatively poorly understood at present, but no discussion of contemporaneous sulphur isotope data.

This is a useful criticism and we have now included the sulphur isotope record in our revised analysis. The d_{34S} record does not show a significant change through the Ediacaran, which is consistent with our model predictions, which we show in new figure 6.

As the reviewer correctly points out, the model ' f_{py} ' value is lower than those indicated in some reconstructions. Models of this type (COPSE, GEOCARBSULF) have generally significant and stable rates of gypsum burial and weathering, and do not switch to a fully-pyrite-dominated system under low O_2 , although in our COPSE simulations, pyrite burial is considerably increased relative to the present. We note that there are significant Neoproterozoic gypsum deposits, and that f_{py} is difficult to calculate accurately (Early Cambrian f_{py} is ~ 0.2 - 0.5 in Wu et al. 2010 and 0.8 - 1 in Canfield and Farquhar 2009) so pyrite burial may not have entirely dominated the real system. We intend to address improved Proterozoic S cycling in future model versions, but we are confident that it does not impact the results of the current study: neither the geologic record nor our model indicate a significant change in pyrite-versus-gypsum deposition over this time, so in that respect our proposed mechanism is consistent with the record.

Third, the way that geochemical proxies are dealt with in Fig. 1 of the main text is cursory to the point of being almost meaningless. What does "Earth system oxygenation" mean? This is fine for the first sentence or two of a caption, but if not elaborated on is not very useful. Selenium isotopes, cerium anomalies, and iron speciation all respond to completely different geochemical conditions, are preserved in different sedimentological archives, and exist along varying degrees in the trajectory toward being well-established geochemical proxies. Even within a single proxy system the figure is frustratingly vague - what does "deep water O_2 " mean, for instance? To what extent does absence of evidence amount to evidence of absence in this regard? This point is particularly important given the implication of the figure that deep water O_2 only emerges after the secular increase in degassing hypothesized by the authors. How well-constrained is this?

We apologise for the cursory presentation in the original version. We have substantially revised our presentation of the geochemical proxies in Figure 1 and have elaborated the caption accordingly. We have taken what was a summary of Fe-speciation data out of the spreading rate panel, and have

created a new figure panel that summarises Fe-speciation results for surface, intermediate and deeper water. We also include in that panel a summary of where redox sensitive trace element (RSE) data suggest intervals of ocean oxygenation. The caption and the main text provide citations to the primary data sources. The caption now explains that Se isotopes should be tracking overall ocean oxygenation, as should RSE enrichments in locally euxinic black shales, Ce anomalies track regional to basin-scale oxygenation, and Fe-speciation tracks local redox state. The caption also notes that whilst Se isotopes and Ce anomalies suggest a trend towards more oxygenated ocean conditions during the Ediacaran period, the Fe-speciation and RSE data just show a series of transient oxygenation events during the Cryogenian and Ediacaran periods. As noted in the main text, when also considering the RSE data into the Cambrian, a case for an increased frequency / predominance of oxygenated conditions becomes more persuasive. We now make it clearer that all of these proxies are tracking ocean oxygenation which we interpret as being the result of atmospheric oxygenation.

Lastly, I can't help but wonder if the manuscript would be more effectively argued and impactful in a longer format (even if only considerably lengthened as a Nature Communications submission, for which I think there is quite a bit of scope given the current brevity?). The model, despite being low-order, is complex, and even though there is little uncertainty analysis in the current manuscript there are already 16 supplementary figures. The basic model framework has been presented elsewhere (Bergman et al., 2004; Lenton et al., 2018), and I appreciate that there are editorial constraints on the short form. But I would argue that the paper would be much more convincing and impactful beyond the relatively small modeling community if more of this were to be made transparent in the main text of the paper. This is particularly true given that the basic result - that increasing CO₂ degassing leads to an increase in atmospheric O₂ - is a rather obvious result based on the way the model is constructed.

We have substantially revised and lengthened our paper to bring more sensitivity analysis and discussion into the main text in a way that is more accessible. The manuscript now contains many aspects that were previously in the SI and a large number of now-redundant SI figures are removed.

We stress that the 'obviousness' of our proposition is not a weakness or simply a product of our model construction. The relationship we describe between overall carbon input and Earth oxygenation is inherent in widely-held views (among Earth scientists) of the operation of the long term carbon cycle. What is interesting is that the Ediacaran results we show represent an example of an O₂ increase that does not relate to a positive $\delta^{13}\text{C}$ excursion. This possibility is something that has been overlooked in the literature and, if correct, will be important for many Earth scientists trying to understand the operation of the carbon cycle, particularly in the Neoproterozoic.

Minor comments:

- It would be helpful if the panels in Fig. 3 were labeled in the figure itself to correspond with the caption text.

Panels are now labelled throughout

- Panels in Fig. 4 should also be labeled. Also, the discussion around Lines 141-145 does not square with the panels shown in the figure itself. Double-check this.

Panels are now labelled throughout and the results section and figures have been revised.

- References 45 and 50 are the same paper (though one is in the bibliography as "in press")

We have amended our referencing here

Reviewer #2 (Remarks to the Author):

This manuscript by Williams et al. presents a new application to the COPSE base model to test whether a change in plate tectonics (i.e. subduction rates) would have an effect on atmospheric oxygen levels. Building on much of this workgroup's prior work, they show that oxygen levels increased throughout the late Neoproterozoic, reaching levels that are thought to be high enough ($pO_2 = \sim 0.1$ PAL) to support the animal life that comprise the Ediacaran fauna. The driver for this oxygenation event relies upon the assumption that increased weathering rates, driven by an increase in degassing of CO_2 from higher subduction rates, delivered more phosphate to the oceans, which elevated primary production and organic burial rates (i.e. a net increase in atmospheric O_2). Though this seems like a long line of assumptions to include in a model, the authors at least acknowledge that parameter constraints during this time are not well known. These model predictions are consistent with other geochemical studies that suggest oxygen levels were increasing during this time, as well as being above the minimum levels needed to sustain metazoan life. Thus, I'm not too surprised when the curve for subduction length (Fig. 1) looks almost identical to the O_2 curve (Fig. 4).

This is a fair assessment of the work, but we have tried to make it clearer in the revision that although this long line of assumptions indeed operates in the model, the mechanism can be explained much more elegantly – a higher rate of \$CO_2\$ input necessitates a higher rate of overall carbon burial, and therefore more burial of organic carbon, which produces \$O_2\$. This contrasts with the more traditional view that increases in \$O_2\$ must be driven by increased burial of organic C relative to carbonates, which would cause a rise in \$d_{13}C\$.

This is a fairly well written manuscript, though it does have some editorial errors (see below) and the modeling appears to be carefully done with appropriate sensitivity testing, but I have some concerns that cause me to recommend that this manuscript not be accepted in its current form. I'm not an expert in the COPSE modeling approach, so I'll defer to the other reviewer's expertise in critically reviewing this model and the inherent assumptions. However, a read through the text and supplement have raised a few concerns that I think the authors should address and state more clearly in the main text.

We note that the sensitivity analysis has now been considerably expanded and improved in response to the other reviewers.

First, I'm a bit surprised that the authors seem to discount the strong control that the sulfur cycle plays on atmospheric O_2 levels via pyrite burial. This version of the model appears to work on the basis that pyrite burial is reciprocal to that of organic burial (I'm probably overstating this, you say balanced by in the text) to regulate atmospheric O_2 levels, a process that the authors take from a paper by Bob Berner written 35 years ago (Berner and Raiswell, 1983). However, it appears that in some of Berner's later work (and many others) that the sulfur cycle plays a much more important role in affecting atmospheric O_2 than just the carbon cycle. For example, in the first version of the

GEOCARBSULF model (Berner, 2006), 1 mol of O₂ is produced for every mol of organic carbon that is buried, but nearly 4 mols of O₂ are produced with one mol of pyrite that is buried (ignoring the organic carbon used during microbial sulfate reduction). Is this wrong? If so, correct this line of reasoning in the text.

The reviewer is correct that mole-for-mole, pyrite burial can produce more oxygen than carbon burial (it is ~2:1 if we consider each mole of S buried as pyrite versus a mole of Carbon buried as C_{org}, which is how these processes are usually modelled). However, the rate of burial of organic C is around an order of magnitude larger than pyrite S burial rates, so dominates O₂ production at the present day. This is true for the GEOCARBSULF and COPSE models, and we have made this clearer in the text.

We have now fully investigated the model sulphur cycle in the main text. The COPSE equation for pyrite burial responds to O₂ levels and organic C availability, so does provide some feedback on O₂ rise, and this formulation has been 'ground-truthed' over the Phanerozoic (Lenton et al., 2018), where the model manages to replicate major shifts in δ³⁴S. Pyrite burial is important in the model, and we now note that ~30% of the O₂ production comes from burial of pyrite in our Ediacaran model run. The revised paper now uses the fully revised COPSE model (Lenton et al. 2018) and thus includes sulphur degassing terms as well as weathering, so the increased degassing increases sulphur input whilst also representing an increased sink of O₂, somewhat counter-balancing the increase in pyrite burial. The proportion of pyrite burial relative to gypsum burial only changes slightly (as now shown in Figure 6) resulting in a stable modelled δ³⁴S, reasonably consistent with the geologic record for the Ediacaran, as we now show in Figure 6D.

In the supplement the authors present their model estimates of the sulfate reservoir size and pyrite burial rates, but these results all seem to be based on the organic burial rate (which is derived from oceanic phosphate concentrations, which are controlled by silicate weathering rates, which are controlled by CO₂ levels, which are controlled by degassing as a function of subduction rates, which are controlled by mid-ocean ridge length). As I stated above I am no expert on the COPSE model, so for folks like me I would like to see the authors explain more clearly why the sulfur cycle isn't as important as they think, or explain why my example above is incorrect. Maybe it's a matter of the sulfate reservoir is much smaller than today?, but I would think that it should still play an important role. It would be helpful to see some model runs where this 'canceling out' of pyrite burial and oxygen release wasn't parameterized in this way (Equation 33). What if pyrite burial contributed XX amount of atmospheric O₂?

As above we have explored the sulphur cycle much more fully in the revised paper, aided by the improved COPSE model that includes S degassing terms – and therefore increases sulfur throughout as well as carbon in response to increased degassing. Atmospheric O₂ rise does act to weaken pyrite burial rates (as in equation 23 (equation 33 in the original SI), which is unchanged), but the increasing sulphate concentration and organic C production outweigh this and pyrite burial does increase in our main experiment (fig 6), although, as noted above, S degassing (an O₂ sink) also increases, damping the net effect of changes of S cycling on the overall oxygenation. Still, we are grateful to the reviewers for encouraging further analysis of the S cycle, as it has led us to properly document that changes in S cycling are an important part of the overall mechanism.

Secondly, and maybe this is my problem and not keeping up with all of the latest papers from this

prolific workgroup and understanding the importance of the COPSE model, but after a few reads of this manuscript and I find myself left thinking that this approach uses more assumptions and guesses at parameter values than known values. I understand that it is impossible to know what uplift rates were like 600 million years ago.... Or today for that matter, but I'm left thinking that the modeled O₂ curve is simply a result of some calculations using constants values/rates on the subduction curve in Figure 1.

This analysis is broadly correct in that we are seeking to highlight in this paper that increasing the overall C (and S) input to the surface Earth system must result in higher overall output fluxes and therefore more burial of organic C and pyrite S (so O₂ release) – but with a roughly equal increase in the burial of the oxidised forms of these species that results in only minor changes in δ¹³C and δ³⁴S. This idea has been hinted at in the past but not explored quantitatively with respect to any actual event.

The basics of this idea could be shown without modelling, but we employ the COPSE model to show that this idea can match the quantitative geochemical records for the Ediacaran. We have improved this model testing, and made the rationale behind it clearer in the revision, including noting that COPSE (although using a number of uncertain parameters) has been tested against multiple geochemical records during the Phanerozoic.

Not to diminish the importance or impact of this modeling work, which I appreciate is difficult to do and assign values to forcing factors, but the modeled O₂ curve is essentially the same shape as the subduction curve with a slight bend during the Ediacaran to account for increased uplift rates associated with the formation of Pannotia. This basically suggests that tectonics controls everything, which I suppose it does, but I suspect there is more to the story than just subduction zone length. I'm not sure I have a good answer for how to placate this critique other than to say I would like to see more effort put into constraining the model at certain time points. For example, the inflection in uplift appears to be made based on sediment abundance at the end Proterozoic, but are there other estimates the authors can draw upon? Perhaps the database hosted on Macrostrat.org (age and rock type) can provide some constraints, or at least be used in sensitivity testing? Surely this would affect the volume of rock being weathered... which should have different average phosphate concentrations. Or, perhaps the rates can be adjusted slightly at different time points to fit the ⁸⁷Sr/⁸⁶Sr curve.

We don't think tectonics controls absolutely everything but we do think it played an important role in the interval we are focusing on, and the paper is best viewed as an attempt to isolate and quantify that role. We do think that a primary driver of atmospheric O₂ increase in the Ediacaran was the increased rate of tectonic recycling, hence the similarity between the curves. In the revision we have made a better attempt at fully-constraining the model parameters through the Ediacaran period. We have taken on board the reviewer's suggestion and used a linear increase in uplift rates that is constrained by the ⁸⁷Sr/⁸⁶Sr curve, and have added a full Monte-Carlo sensitivity analysis to uncertainties pointed out by reviewer 3. Overall, we are happy that the model produces a reasonable long-term fit to major variations in global biogeochemistry through the Ediacaran.

My relatively uninformed read of the model description is that a few rates/constraints were chosen from previous work, and because the model estimates kind of match geochemical data measured in

other studies (e.g. Fig. 3), it's close enough. Perhaps if this 'ground truthing' the model isn't possible, then I feel that the authors should more clearly state the inherent assumptions in this approach.

The revised COPSE model has been 'ground-truthed' across the whole Phanerozoic by comparison of model predictions for ocean and atmosphere chemical composition and $\delta^{13}\text{C}$, $\delta^{34}\text{S}$ and $87\text{Sr}/86\text{Sr}$ isotopes (see Lenton et al. 2018 for a summary). We now use this full model in the revised paper (it was not published when the original paper was written) and have noted this ground-truthing explicitly in the text.

It wasn't clear to me in my first read that this model assumes a lot, including: a constant supply of phosphate, constant weathering rates, a constant mix of rock with constant phosphate contents, phosphate is the only nutrient needed to produce a constant amount of primary productivity, and constant organic burial rates uniformly in the oceans. I have no evidence to say whether this is wrong or inaccurate, but I think it'd be better to at least let the reader know how many assumptions are made with this approach.

All of these things do in fact vary in the model. We are sorry for the confusion and have amended any text that may have mislead here. Overall we are taking a fully-dynamic predictive model that produces a reasonable prediction of Phanerozoic climate and are running it for the Ediacaran subject to an increase in degassing rates and erosion rates. The model helps us quantify any expected O_2 rise due to C degassing, lets us see how other uncertainties affect the likelihood of this, and lets us see if the mechanism is falsified by isotopic tracers such as $\delta^{13}\text{C}$ or $\delta^{34}\text{S}$ (both of these show no clear trend in the geological record so if COPSE produced a strong upturn/downturn in $\delta^{13}\text{C}$ or $\delta^{34}\text{S}$ we would know our mechanism was incorrect).

We have made the reasoning and workflow more clear in the revision and hope this fixes the problems pointed out by the reviewer.

That said I found some other minor issues with the manuscript that require some attention spelled out below. I find that this manuscript has potential to be an important contribution to the field, particularly if the authors can underscore the major assumptions that the COPSE model is based on and that it isn't really constrained by highly resolved measurements.

- Line 9: Please use the ages from most recent Geologic Timescale, it's 2018 after all. The end of the Cambrian is 485.4 Ma. Also check that the ages for the Neoproterozoic periods are those in the 2012 GTS.

We have changed the end-point of the Cambrian to 485.4 Ma.

- This may be a style thing, but when discussing time, one should use the word 'during' instead of 'in'. For example, the first sentence of the abstract should end as "~850 to 541 Ma during the Neoproterozoic Era." This issue is present throughout the manuscript.

This has been changed throughout the manuscript.

- Figure 1: This figure is incomprehensible without some more explanation in the text, figure caption, and/or figure itself. It's a bit odd that Fig. 1B is explained in the main text before Fig. 1A, but what is more puzzling is what the Se and Ce data indicate. Which end of this figure is more oxic or anoxic?

What does the dashed blue line mean? What is the does the “Deep water O₂” mean, from who’s study, and why is this plotted on the figure showing subduction zone length? What does the gray area mean around the black line in 1A? A little annotation to the figure and caption will go a long way to help the reader understand what is to be communicated here.

Figure 1 has now been expanded and accompanied by a more informative caption, which explains what the Se and Ce data indicate and that oxygenation is upwards (as well as explaining what the dashed blue line is). What was a summary of Fe-speciation data for deep water O₂ on the first panel has now been removed. Instead we have added a panel giving a more detailed summary of Fe-speciation results for shallow, intermediate, and deeper locations, and also included in this panel a summary of redox sensitive trace element (RSE) data for intervals of ocean oxygenation.

- Line 53: Please explain the last part of this sentence. Why does organic burial only temporarily increase atmospheric O₂? I cannot follow.

Here we are surmising a previously postulated mechanism for pO₂ rise at the Neoproterozoic/Phanerozoic transition whereby an enhanced nutrient flux to the ocean as a result of the Snowball Earth glaciations could enhance organic carbon burial and thus increase atmospheric O₂. However, we would not expect these pulses of nutrients to continue indefinitely in order to instigate a step-change in atmospheric O₂ – eventually the pulse would end and the system would return to the previous state. We have noted this in the paper.

- Line 58: A transition is needed here, or perhaps introduce this notion earlier in this paragraph.

We have added a better transition here

- Line 63: An alternative what?

We have changed this to make clearer that we are presenting an alternative mechanism by which an oxygenation could have been driven during the late-Neoproterozoic.

- Line 64: Can you include these data in Fig. 1A? It seems like this would strengthen your argument that geologic data and modeling results are consistent with each other, which makes it seem reasonable that subduction zone length can be used as a proxy in other ways.

We now show the McKenzie data in figure 1.

- Lines 78–79: Late and early should be lower case. These are not formal time intervals (unlike the Early Ordovician, for example). Please make the appropriate changes throughout the rest of the text.

We have made this change

- Line 86: Seems like ‘protected’ is an odd word choice.

We have changed ‘protected’ to ‘shielded’ which is a more appropriate word choice.

- Lines 94–98: This assumes fractionation is constant, that there isn’t a fractionation effect associated with carbonate formation, and that there is complete oxidation of organic matter.

We have added this further description to the caption for figure 2 as we could not find a way to incorporate these caveats here without breaking the flow of the paragraph.

- Line 108: What effect?

We refer to the effect that changing rates of organic carbon and carbonate burial have on $\delta^{13}\text{C}$, noted on the previous line. We have now merged these paragraphs to make this clear.

- Line 109: Might be a good place to remind the reader what reduced source you're referring to, especially if sulfur isn't as important in this model.

We now write 'carbon burial rates'

- Line 112: What does 'sped up' mean? Doubling? Faster burial and weathering rates?

This refers to an increase in the rate of weathering and burial of organic carbon relative to carbonate carbon (which would force a step change in the $\delta^{13}\text{C}$ record).

- Line 115: Does this mean that atmospheric CO₂ levels were constant? I guess not as this figure is buried in your supplement. I would suggest you promote this figure (and a sulfur reservoir/pyrite burial plot) to the main text. Do these values correspond to other CO₂ estimates? How do they agree with the timing of the glaciations during this interval? I thought that the way we got out of these glaciations was because CO₂ levels increased greatly during this time, but you show there's a steady decrease. Something is off here and you should probably acknowledge this discrepancy.

CO₂ levels were not constant, but carbon inputs and outputs must balance over long timescales as the residence time of surface carbon is very small when compared to geological time.

We have added modelled CO₂ and sulfur cycling to the main figure. We see a rise in modelled CO₂ from ~14 to ~16 PAL, before a slight decrease to ~15 PAL going into the Cambrian. This is consistent with previous modelling (as shown in Mills et al. 2017, Nature Comms) and with a general increase in surface temperature between the cold Cryogenian and Warm Cambrian due to the rising solar flux and increased degassing rates (similar CO₂ over 100 Myrs equates to warming). The steady decrease in CO₂ was only observed in the 'uplift only' model we previously plotted in the SI. Admittedly the SI was a bit confusing and we have streamlined this.

- Line 142: Are these the correct figure references?

We have revised our figures and corrected the references

- Line 143: Missing superscripts here.

We have amended this

- Lines 158–160: Wait.... How can pyrite burial be suppressed by rising O₂ levels (re-oxidation?) but organic burial can occur unchecked? It seems to me that burial means burial and that either or both can occur without later re-oxidation. As stated above, doesn't pyrite burial yield 4 times as much O₂ compared to organic burial? It seems like this might be a good place to differentiate the two models and why GEOCARBSULF and COPSE are so different with respect to how they deal with the sulfur cycle.

This section has been replaced with more complete results and discussion text. Pyrite burial suppression by rising O₂ represents a limitation on rates of microbial sulphate reduction. Neither the carbon or sulphur system has an explicit representation of reoxidation. This formulation in COPSE is admittedly simple but produces reasonable d₃₄S results over the Phanerozoic. GEOCARBSULF calculates pyrite burial rates from the d₃₄S record and the total sulphur throughout, and given a rise in degassing and near-static d₃₄S, should predict a rise in pyrite burial rates as we show in our model. A GEOCARBSULF-based reconstruction of this period would be interesting and is something we are working towards.

- Figure 4: Again, this figure needs better annotation. There aren't letters associated with the different panels. Why show the data from McArthur et al. but refer to the data from Cox et al. in the main text? Why not both? Superscripts missing for ⁸⁷Sr/⁸⁶Sr and the wrong symbol is used in 'δ¹³Ccarb' in the caption.

We have fully revised this figure.

- Lines 210–213: This is a fairly subjective claim to make that measured δ¹³C values don't change throughout this interval. δ¹³C values are all over the place and rarely at 0‰. One could argue that there are several baselines that are achieved (+4‰ from 700–650 Ma, -1‰ from 650–610 Ma, +4‰ from 610–580 Ma, etc.). Could these shifting baselines reflect short-term changes in weathering, degassing, or organic burial rates? Do you actually calculate the average value over this time and find that it is 0‰ to agree with your model? I'd be surprised if the average value isn't more like +2‰. What would it mean in your model (i.e. sensitivity testing) if the average δ¹³C value was -2 or +2‰? How might that affect organic burial rates and ultimately O₂ levels? Ultimately I think it's too much of a simplification to say that the long-term δ¹³C average is 0‰, particularly when these excursions are thought by many to reflect major changes to the redox state of the oceans.

We have been clearer to focus this study on the Ediacaran and now show statistics for Ediacaran d₁₃C, which indicate no clear trend and that the average value is close to 0‰. It is certainly true that the positive and negative swings during this period may relate to global processes, and probably to ocean oxygenation. We have more carefully stated that the focus of our paper is on a steady rise in atmospheric O₂ over this period and whether or not that is accompanied by a secular trend in long term d₁₃C, rather than seeking to explain each excursion in the record.

- Line 231: EdiacaraN?

We have amended this.

Reviewer #3 (Remarks to the Author):

Review of “A Tectonically Driven Ediacaran Oxygenation Event” by Williams, Mills, and Lenton

In this study, the COPSE model is used to test a new hypothesis for the Neoproterozoic Oxygenation Event (NOE). The paper argues that a ~1.7 fold increase in crustal production between 720 and 540 Ma drove an increase in carbon dioxide outgassing, and that this necessitated greater total carbon burial to balance the carbon cycle. Although the proportion of carbon buried as organic carbon did not necessarily change, elevated total burial resulted in greater organic burial, and it is the absolute organic burial flux that determines the oxygen source flux. The COPSE model outputs suggest that this increase in the oxygen flux was sufficient to approximately double the atmospheric O₂ content, which could have enabled the appearance and diversification of complex animals in the Cambrian. It is also argued that their COPSE model outputs track the observed Sr isotope record, although it is necessary to impose a sizeable change in erosion/uplift to achieve this.

This is an interesting study that presents a novel hypothesis to explain the NOE. The paper is commendable for testing the self-consistency of their idea with a complete biogeochemical model rather than merely presenting back-of-the-envelope calculations that can overlook important feedbacks and constraints. The paper is generally easy to follow and, with a few minor exceptions that could be easily remedied (see below), the model is described in sufficient detail for other researchers to try and reproduce the general findings.

However, I cannot recommend publication at present for several reasons enumerated as follows.

1) The main issue I have with this paper is that there are a large number of parameters, parameterizations, and coefficients that are not well constrained so that the conclusions may not be accurate and robust. The paper claims (line 255) that “our predicted long-term Ediacaran oxygenation event driven by increased tectonic degassing is robust to model uncertainties”. This has not been demonstrated in the paper. There should be some sort of sensitivity test with key unknown parameters - ideally some kind of Monte Carlo analysis that runs the model thousands of times, sampling a broad range of plausible parameter values.

In the revision we have run a large (10,000 runs) Monte Carlo analysis over the most important parameters affecting the potential for tectonically-driven Ediacaran oxygenation. This follows the procedure of Royer et al. (2014) who used a similar analysis of the GEOCARBSULF model in the Phanerozoic. Whilst there are indeed many model feedbacks that can dampen the O₂ rise, to actually counter it would require either assuming that the degassing rate declines or adding in another external forcing or dynamic parameter change that opposes O₂ rise itself. Thus we do believe that our qualitative result is robust to any model uncertainty. The quantitative result has however been better constrained by following the advice of the reviewer and we are grateful for this.

Alternative parameterization functions should also be considered. In the paper abstract it is claimed that pO₂ doubled during the Ediacaran. The paper needs to show that this quantitative statement holds true for a wide range of model assumptions, and not just for the somewhat arbitrary parameterizations and parameter values adopted in their nominal model. To offer some concrete examples, there are two questionable model assumptions that could, if modified, potentially change the paper’s conclusions about pO₂.

As above the new analysis has allowed us to better quantify the expected O₂ rise, this is indeed dampened and is now around a 50% increase on average, but of course the parameter space allows for larger or smaller increases. As the reviewer has recognised, the main point of the paper is introducing a qualitative mechanism for O₂ rise that does not significantly impact δ¹³C or δ³⁴S.

2a) The first questionable assumption relates to the parameterization of seafloor weathering (equations 30a and 30b in the supplementary material). This parameterization implies seafloor weathering is purely temperature dependent, with no dependence on crustal production or ocean pH. The pH dependence is arguably small enough to be neglected, but the crustal production dependence is not. If subduction zone length increased in the Neoproterozoic as the authors claim, then there would presumably be more seafloor basalt available for weathering, and therefore increased uptake of CO₂ as carbonate in the seafloor. Indeed, many previous carbon cycle models have included a linear dependence on crustal production or spreading rate (e.g. Sleep and Zahnle 2001, JGR Res; Franck et al. 2002, Tellus B; Le Hir et al. 2008, Geology; Foley 2015, ApJ) or a power law dependence that encompasses linear proportionality (Krissansen-Totton et al. 2018 PNAS). This is potentially important because if the actual response to elevated degassing is increased carbonate uptake in the seafloor, then the proportion of carbon buried as organic carbon may drop, and the rise in oxygen may be diminished. For example, the paper's initial (preset day) value for seafloor weathering is 1.75 Tmol C/yr. If this increased by ~1.7x during the Ediacaran due to a linear dependence on crustal production, then the final seafloor weathering flux (before accounting for T-dependence) will be 1.75*1.7 = 3.0 Tmol C/yr. Thus there is a potential ~1.2 T mol C/yr increase in carbonate burial in the seafloor that could offset the increase in organic burial.

We regret that the SI equations for seafloor weathering did not include the relationship with crustal production. This mistake occurred when shortening thesis work (which tested both equations) to write the paper, and we apologise. As with the previous COPSE versions that include this flux (Mills et al., 2014a,b), the model does indeed use a linear dependence of seafloor weathering on crustal production. We now use the published 'COPSE reloaded' model (Lenton et al. 2018) for all analysis and have amended the SI accordingly.

2b) This effect is compounded by the paper's choice of a relatively weak temperature dependence for seafloor weathering (equation 30a). Coogan and Dosso (2015, EPSL), and Krissansen-Totton and Catling (2017, Nat. Comm.) both suggest that stronger temperature dependencies are possible. This would also have the effect of producing a large seafloor weathering response to an increase in degassing which would act to offset the organic burial increase.

This is a great suggestion and we have tested the full range of temperature dependencies from Krissansen-Totton and Catling in our revision. This does work as the reviewer suggests.

2c) Furthermore, differing seafloor weathering parameterizations will also affect the Sr isotope model. The model fit to the Sr isotope record could potentially be worsened by including a crustal production dependence and a stronger temperature dependence for seafloor weathering. This should be investigated.

Again this is a useful suggestion, and operates as the reviewer describes. The fit to Sr is indeed worsened and the model now requires a larger (although still plausible) increase in uplift rates to match the Sr record, dampening O₂ rise and producing a fuller test of the model robustness overall.

3) The second questionable model assumption I noticed was the choice to ignore the increased flux

of reduced gases associated with elevated carbon dioxide degassing. For the same mantle redox conditions, higher CO₂ outgassing will also result in a greater flux of reduced gases such as H₂ and CO, which are oxygen sinks and act to offset the pO₂ increase caused by the higher throughput of carbon. The paper acknowledges this possible sink on lines 225-228 of the supplementary materials, but then argues that it is unimportant because the reduced gas flux is much smaller than organic weathering and other oxygen sinks. Canfield (2005, Annual Review EPS) is cited to support this claim, although I could not find specific numbers in this Canfield paper. Canfield (2006, Phil. Trans. Royal Soc. B) estimates the modern H₂ flux as 1.8-5.0e11 mol/yr, which is indeed small compared to the 7.75e12 mol/yr present day oxidative weathering flux assumed by the authors. However, the H₂ flux from Canfield (2006) is at the low end of estimates from the literature. Other estimates for modern H₂ outgassing range from 0.9e12 mol/yr to 2.2e12 mol/yr (see Holland 2009 GCA; Catling and Kasting 2017, Atmospheric Evolution on Inhabited and Lifeless Worlds, p. 208-210). If H₂ outgassing were at the upper end of this range and responded linearly to total degassing then the O₂ sink would increase by $2.2 \times 1.7 - 2.2 = 1.5$ Tmol/yr over the Ediacaran. This would also dampen the rise in oxygen somewhat.

This is an excellent point. As the reviewer notes, our choice to ignore this possible dynamic was based on low estimates, and on the fact that it is not part of the COPSE model (which has never included direct mantle input). In the revision, we have extended the COPSE model to include a 'ridge degassing' flux of both CO₂ and H₂, in addition to the crustal recycling fluxes. A ridge CO₂ flux was required to balance the coupled C and O cycles because the ridge O₂ sink must be compensated for by reducing the oxidative weathering rate at present day, leaving a gap in our carbon budget. We feel this is reasonable overall. This setup roughly follows Hayes and Waldbauer (2006).

4a) Finally, I wonder to what extent the paper's choices of absolute present day fluxes affects the conclusions. Although it is not plotted or stated anywhere, I tentatively infer that the assumed modern day value for carbonate weathering used in this study is 6.25 Tmol/yr (based on steady state values in page 6 in supplement). Estimates in the literature range from 7-14 Tmol/yr (Hartmann et al. 2009, Glob. Planet. Change). The same paper lists a wide range of estimates for silicate weathering. The present manuscript also adopted a modern total outgassing flux of 7.9 Tmol/yr, whereas literature estimates range from 4-10 Tmol/yr (Lee and Lackey 2015, Elements; Berner 2004, The Phanerozoic Carbon Cycle). Similar uncertainties exist for the organic carbon fluxes, although I have not dived into the literature on this. The submitted manuscript should test the sensitivity of the results to changing present day fluxes within plausible bounds.

This is a good point and we have tested the choice of present day fluxes in our improved Monte Carlo analysis (see table 1). We have switched to the updated COPSE model (Lenton et al. 2018) for our revisions, which has updated present-day fluxes based on a careful literature review.

4b) A related issue is whether it is reasonable to assume carbonate and organic carbon degassing increases linearly with subduction zone length. Intraplate volcanism on the modern Earth is potentially quite large (Lee and Lackey 2015, Elements) and so total degassing may respond less than linearly to elevated rates of subduction. This would potentially lessen the change in carbon throughout and therefore lessen the Ediacaran rise of oxygen.

This is a fair point, but it is also likely that total C degassing may respond more than linearly to elevated rates of subduction, through thermal decomposition of crustal carbon at continental arcs (see other works by Cin-Ty Lee and Ryan McKenzie for example). Thus we think the linear

relationship is a fair middle ground.

To be clear, I'm not saying the proposed oxygenation mechanism doesn't work. The paper provides a scenario that might explain both the Neoproterozoic rise of oxygen and several other geochemical data sets. However, it's not enough to merely show such a scenario is possible. The paper needs to demonstrate that these conclusions are robust. Specifically, if a more standard seafloor weathering parameterization is adopted, if elevated H₂ fluxes are accounted for, and if broader ranges of present day fluxes are considered, do the paper's conclusions about oxygen still stand? The paper could be made more convincing and impactful by demonstrating that an approximate doubling of pO₂ occurs despite these uncertainties.

We are very grateful to the reviewer for these suggestions. The resulting uncertainty analysis has strengthened our paper and more clearly shows our confidence in the operation of this mechanism. The average O₂ rise is dampened from ~2 fold to ~1.5 fold, but is still significant.

Other important issues:

5) How exactly is pO₂ through time being calculated? As far as I could see there is no differential equation that specifies the time-evolution of the atmospheric oxygen reservoir. The reader has to piece it together from Figure 2 and the various carbon flux equations. Please explain clearly how the atmospheric oxygen reservoir is calculated since this is crucial to the paper's main conclusions. Also, please explain the k_{O₂} term in equation 37 in the supplementary material.

We have updated the supporting information to clear display all important calculations. These are also explained in further detail in Lenton et al 2018 (Earth-Science Reviews), and we make it clear that we are using that model as a baseline.

6) Lines 50-61. I think this paragraph is too dismissive of previously proposed mechanisms for the NOE. The paper argues that all these mechanisms imply a rise in d¹³C of carbonates, and that such a rise is not observed in the isotope record. However, Krissansen-Totton et al. (2015, AJS) found that there is a statistically significant increase in both d¹³C(carbonates) and steady state fractional organic carbon burial between the mid Proterozoic and Phanerozoic. The magnitude of this fractional organic burial increase is around 30%, albeit with large uncertainties. A 30% increase in fractional organic burial is comparable to the increase in the paper's model (Fig. S15A), which is enough to cause an approximate doubling in pO₂. This increase in steady state d¹³C is not visible in the 550-700 Ma record plotted in this study because the broader context of the time series is omitted. Perhaps the observed increase in d¹³C is insufficient for the alternative NOE hypothesis summarized in lines 50-61, but it is inaccurate to say there has been no change in d¹³C and imply that the other proposed mechanisms that predict d¹³C increases do not work. Other possibilities for decoupling organic burial and the d¹³C record have also been proposed and ought to be mentioned (e.g. Schrag et al. 2013, Science).

We agree that there is a change in carbonate d¹³C between the mid Proterozoic and Phanerozoic, and do not dispute the whole-Earth-history trajectories shown by Krissansen-Totton et al. But the timeframe of interest for this paper is the Ediacaran period, and the paragraph in question includes only the Ediacaran and broader Neoproterozoic. We now show C isotopes in the SI and calculate trends over the Ediacaran period and the Neoproterozoic Era. There is no long term trend over the Ediacaran and there is a gradual negative trend over the Neoproterozoic. Thus we think it is fair to say that there is no clear rise in d¹³C over either of these timeframes. The Schrag et al. mechanism is

critiqued recently by Shields and Mills (2017, PNAS) and we do not think it is particularly useful to repeat that criticism so soon without response.

7) The incorrect claim that the $\delta^{13}\text{C}$ record has been constant is repeated on line 112 and again on lines 145. Relatively modest changes in fractional organic burial can cause substantial changes in $p\text{O}_2$, and such changes are consistent with the carbon isotope record. Statements about the constancy of the carbon isotope record should be adjusted accordingly.

We stand by the statement that there is no whole-Neoproterozoic or whole-Ediacaran rise in $\delta^{13}\text{C}$, and have now shown this statistically. The later statement that implied general stability over the whole of Earth history has been revised as we agree with the reviewer here.

8) The descriptions of the degassing forcing (Supporting Information 3) and the uplift forcing (Supporting Information 4) are somewhat confusing. For instance, the paper states that 100 Monte Carlo simulations were run, and the 1-sigma and 2-sigma standard deviations were calculated. However, Fig. S5 shows 20% error bounds, not 1-sigma or 2-sigma errors. If the dashed lines are 20% errors, this seems to suggest the 2-sigma envelope is quite large and potentially consistent with constant relative degassing. Is this interpretation correct? If so, this should be clearly stated in the main text.

The '20% error' bounds were a precursor to using the actual uncertainty calculated in the degassing forcing of Mills et al (2017) and were simply that percent of the forcing value, as used by Royer et al. (2014) in their GEOCARBSULF study. The 1 and 2 sigma boundaries were the result of the Monte-Carlo simulations run between these bounds.

We have substantially revised the Monte-Carlo analysis and its presentation and now use the error window from Mills et al. (2017) for degassing and prescribe a linear increase in uplift in line with the Sr record, as suggested by reviewer 2.

9) Similarly for the uplift forcing, the plotted 20% errors suggest that the 2-sigma envelope is very large. I also don't understand how the 750-550 Ma portion of the curve is derived from (or consistent with?) the scarce supercontinent and orogeny constraints in this time span.

Following suggestions by Reviewer 2, we now define an uplift curve based on the model fit to the $^{87}\text{Sr}/^{86}\text{Sr}$ record, which is in line with estimates for Ediacaran vs Cambrian sediment abundance. We also present results in the SI under a constant uplift scenario.

10) More generally, what is the purpose of these error calculations anyway? It looks like only the median degassing and uplift curves are being used as input for the COPSE model calculations.

We have improved the Monte Carlo analysis as previously stated in response to prior comments. We use the subduction zone length data from Mills et al (2017), in which an upper and lower estimate are provided for each time point. These data are resampled to every 10 Myr, and at each time step we choose a random value between the upper and lower values for that given time point. Uplift follows the prescribed line.

Minor issues/corrections

11) The specified range for Proterozoic oxygen on line 27 is a little narrow (compare Lyons et al. 2014, Nature).

We have changed this range to 10^{-4} - 10^{-1} PAL based on Lyons et al (2014) Figure 1.

12) Please explain all lines, symbols and shading in the Figure 1 legend. In general, figure legends could be more descriptive and would help improve the clarity of the manuscript.

The Figure 1 legend has been considerably expanded to fully explain the expanded content.

13) Line 142 references Fig. 4c when discussing fractional organic burial, but Fig. 4c is a plot of Sr isotopes.

We have made sure our main text refers to the correct figures/sub-figures

14) Line 205: Refer the reader to the supplementary materials that explain where the uplift curve comes from. On first reading it isn't clear why this uplift curve should be believed.

We have changed the uplift curve based on the model fit to the $^{87}\text{Sr}/^{86}\text{Sr}$ record, and we now explain this in the main text to make sure this is clear to the reader. We note that the uplift change is not required to increase O₂ and actually works against the proposed oxygenation mechanism. Thus we show the mechanism works in spite of proposed increases in uplift.

15) Supplementary material: there are many equations for plant assisted weathering, organic burial on land, vegetation feedbacks etc. I assume these equations aren't used for the version of model adopted in this paper since there was no vegetation in the Precambrian. Perhaps they could be omitted so as not to overwhelm and confuse the reader with unnecessary equations (i.e. only include the pre-plant versions of these equations)?

Following this suggestion, we have removed all equations unnecessary for this study from the supporting information.

16) Supplementary material lines 14-16: It might be worth citing Coogan and Dosson (2015, EPSL) here to justify the temperature dependence of seafloor weathering.

We have fully revised our supplementary information and explain our choice of temperature dependence of seafloor weathering in the main text, and this is now a range of values based on those presented in Krisanssen-Totton and Catling (2017). The values used are displayed in Table 1.

17) Supplementary equations lines 48 and 49: could you write out these weathering equations to remove any possible ambiguity?

The supporting information has been thoroughly revised, with equations now following Lenton et al 2018 (Earth-Science Reviews). We hope that the new supporting information is clearer and easier to follow.

18) Supplementary equations 24, 25, and 27: PG appears to lack a definition (unless missed).

We have added a table of forcing factors to the supporting information in which these factors are defined, and the values given. The PG forcing refers to the paleogeography effect on runoff and weathering, and is set constant at 1 in this study. This is now made explicit in the SI.

19) Supplementary lines 81-83 simply repeat supplementary lines 73-75.

This has been fixed in our SI revision.

20) Fig S3 and S4: Please adjust all large-numbered contours to be in exponent form. Nobody likes counting zeros.

We have moved the steady state analysis to the main text and have labelled the data in a more understandable manner.

21) Why is the organic carbon burial curve in Fig. S15 different to that in Fig. S8? The difference between modeled degassing and dynamic degassing isn't explained clearly anywhere (I searched "dynamic" and it is only mentioned in the supplementary figures). This is another area where more descriptive figure captions would be helpful.

Those figures used different time scales but did show the same curve. We have revised the entire SI to be clearer.

22) Finally, if feasible, the authors should make the version of COPSE used for this study publicly available and provide a link to its location in the manuscript. This would greatly improve the transparency and reproducibility of their work.

We have included a link to download the full 'COPSE Reloaded' model which we use.

Reviewers' comments:

Reviewer #1 (Remarks to the Author):

The authors have made a good faith effort to address my primary criticisms in the first round of review, and although I do not think I agree on every interpretive point my criticisms of the revised manuscript would be largely either nit-picking or moving the goalposts. In particular, the sensitivity analysis is much more compelling, and the explicit incorporation of the S cycle and S isotopes into the discussion represents a significant improvement in my view.

I would be happy to see the manuscript published subject to editorial discretion.

Reviewer #2 (Remarks to the Author):

This revised manuscript by Williams et al. is significantly improved and more measured with respect to their conclusions and interpretations of the modeling. The addition of the uncertainty envelopes is a much needed element to this work, and their approach using the Monte Carlo simulations seems to be a reasonable method given a similar approach is used in other recent modeling papers. This revision also includes several improvements to the explanation of the model, its parameters, and assumptions inherent in this approach, so I commend the authors for their hard work in this revision. The authors seem to have also done an adequate job of addressing all of the reviewer's comments/concerns, as well as improving the figures and captions, which have only helped to strengthen and focus the discussion.

Overall I found a few areas in this revision that I think the authors should reconsider or change before this manuscript can be accepted for publication. Most of my comments are minor (see below), but one major issue that I am still unclear on is in regards to the balance between subduction and destruction/formation of carbonates and organic matter. I may have missed this in the discussion, and if so it might be good to highlight it again elsewhere in the text, but it seems to me that the authors invoke an increase in degassing rates as a function of subduction, but that this subduction does not increase the amount of carbonate/organic matter that is also subducted. The model requires that if CO₂ from degassing increases, that there is a net increase in C burial in equal carbonate and organic carbon proportions to balance CO₂ levels, but increasing overall atmospheric O₂ levels from higher overall organic burial rates. But what happens to these rocks? Does the model require that these rocks are formed but that they are kept from being subducted and ultimately recycled? Nowhere did I find in the text where in the ocean basin this organic burial occurs, so one has to assume that it can be equally distributed in the global ocean (I suspect most is concentrated

along continental margins, but these margins cannot hold an infinite amount of sediment), but some of it most certainly occurs in the deep ocean. If so, some of this sediment would eventually be subducted, thus diminishing the net amount of oxygen produced from organic burial. Again, unless I missed something, this I think is a critical component to this story, potentially a major flaw, and something that the authors need to address still.

A few minor issues:

Line 49 – Something is missing in this sentence. Perhaps change it to “concentrations”?

Line 58 – A reference is needed here for the temperature estimate of Cambrian seas.

Lines 75–109 – Please be consistent with the use of ocean-atmosphere system. For example, Lines 77 and 81 use ocean-atmosphere (which I prefer), but lines 87 and 100 are atmosphere-ocean. I am griping here only because I was once corrected on this same point and understand now that the ocean component is more important than the atmosphere.

Line 93-94 – By “sped up”, do you mean that there was a higher flux of organic burial or increased organic burial rates?

Line 177 – Seems like it would be helpful to clarify that this oxygenation is ultimately accomplished by pyrite burial, as a function of increased sulphate reduction.

Line 529 – Avoid the ‘isotope slang’ of lighter and replace with less positive.

Lines 534–537 – I’m confused by this statement, which invokes euxinia and oxygenation all at once. A little clarification would help.

Line 553 – In the figure you might consider adding “ $\delta^{13}\text{C} = \#\#$ ” under each reservoir just to be crystal clear.

Line 572 – Because every other panel has units associated with the values, you might as well add ‰ after $\delta^{13}\text{C}_{\text{carb}}$ and $\delta^{34}\text{S}_{\text{seawater}}$.

Line 579 – al should be “al.”

Line 581 – Table 1 should be capitalized.

Reviewer #3 (Remarks to the Author):

Second Review of “A Tectonically Driven Ediacaran Oxygenation Event” by Williams, Mills, and Lenton

The revised manuscript is much improved. Most of my comments have been addressed, and the new Monte Carlo analysis is close to what I had in mind when I suggested including an uncertainty analysis in my first review. The authors’ thoughtful responses and thorough revisions are appreciated. With that said, there are still a few remaining issues that I would like to see addressed before this paper is accepted for publication:

Firstly, I don’t fully understand how reduced gas fluxes were incorporated into the model. Table 1 seems to indicate a modern range for the H_2 (?) flux of 0 to 2.2 Tmol (H_2 ?)/yr. To ensure modern day carbon fluxes balance, an equal ridge degassing flux of CO_2 was added to compensate for the reduction in oxidative weathering required to balance the oxygen cycle. This seems reasonable. However, I am unsure as to the functional form of “rgf”. I would assume that the functional form is something like $\text{rgf} = \text{modern_H}_2\text{_degassing} * D$ where D is relative subduction zone length. Is this true? If so, then this should be clearly stated and explained in the supplement. If some alternative parameterization was used, then this should be justified. I don’t think it would be reasonable to sample the modern range of ridge degassing fluxes without any relationship to ridge length; the whole point of my original comment on this subject was that the H_2 flux would rise with total CO_2 degassing, partially compensating for the enhanced carbon throughput. I also don’t understand what is meant by “sedimentary reservoirs being equated to 0 with the addition of a ridge has flux” – please clarify what this means and why it has been done.

Second, although the Monte Carlo analysis in the revised manuscript is an improvement over the original analysis, I have a few additional recommendations. I appreciate that the uncertain parameters I highlighted in my first review have been broadly sampled (Table 1). This makes the paper's key results much more convincing. However, the scope of the Monte Carlo analysis is somewhat limited in that only 5 parameters were varied. I suspect that these parameters are among the most important in controlling the magnitude of the oxygen rise—hence why I highlighted them in my original review—but they probably aren't the only important parameters. The COPSE model has many other parameters and parameterizations, and uncertainty in these could affect the paper's quantitative conclusions (Reviewer 1 clearly shares similar concerns). The overall result that oxygen increased by ~50% is probably relatively robust to these uncertainties. However, I am less certain that the absolute size of the oxygen reservoir would remain around 0.2-0.3 if uncertainties in other parameterizations were taken into account. This is important because in the paper's discussion on animal O₂ requirements, it is claimed that the model predicts a transition from O₂ levels below 0.25 PAL to above 0.25 PAL in the Ediacaran, which is argued to be consistent with crossing the minimum oxygen requirement threshold for the diversification of life in the Cambrian. I am concerned that if the Monte Carlo analysis was expanded to account for uncertainties in other key parameters, then oxygen levels may often remain well below (or well above) 0.25 PAL throughout the entire Ediacaran, and that the number of model runs where <0.25 PAL to >0.25 PAL transitions occurs could be rare.

To offer some concrete examples, I suspect the initial pO₂ value at 635 Ma depends on parameterizations for oxidative weathering (eq 18 in the supplement) and organic carbon burial (equations 6 and 7 in the supplement). Parameterizations of the phosphorus and sulfur cycles will similarly affect the starting pO₂ values at 625 Ma. Additionally, absolute steady state pCO₂ will depend on the temperature-dependence and CO₂-dependence of granite and basalt weathering (equations 14 and 15 in the supplement, and the climate sensitivity in equation 30). Uncertainties in these carbon cycle parameters will, in turn, affect absolute oxygen reservoirs. In some respects, the statement that pO₂ probably increased from <0.25 PAL to >0.25 PAL is equivalent to stating that pCO₂ remained bounded between 12-18 PAL throughout the Ediacaran (Fig. 5B and 5C), which is quite a strong statement that may not be justifiable with an uncertainty analysis varying only 5 parameters.

There are several ways this could be remedied. Ideally, the authors should do a more comprehensive Monte Carlo analysis to investigate the sensitivity of their results to varying a larger set of parameters. If the range in absolute oxygen levels is not strongly affected by this analysis, then the paper's conclusions would be strengthened and the implications for the Cambrian diversification would become even more compelling. Alternatively, if the authors do not wish to expand their Monte Carlo analysis, they should at least highlight the limitations of their uncertainty analysis in the main text and abstract. For example, these limitations should be noted when discussing absolute oxygen levels in the "Animal O₂ requirements" discussion.

Additional comments:

Lines 94-95 and Supporting Information 1. Somewhere in here it would be helpful to summarize some version of your response to my comment (6) in my original review i.e. point out that even though there is probably a step change in relative organic burial from the mid Proterozoic to Phanerozoic, within the Neoproterozoic itself there is no strong trend.

Lines 183-185. It's not entirely clear where this linear increase in uplift comes from. Here, it makes it seem like it is derived from sedimentary records (is this Fig. 4 in reference 42?), but elsewhere (line 32 in the supplement) it seems like it is chosen to fit the Sr isotope record. The response to reviewers makes it seem like the Sr isotope record is being used, but this should be clarified in the main text.

Lines 203-204: Does this mean that the degassing forcing is varying stochastically and dramatically about the median curve in Fig. 1A every 10 Myrs? Would it make more sense to use an ensemble of smoothly varying outgassing curves because stochastic variation on such short timescales is likely unphysical? This probably doesn't make too much difference for the final figures (Fig. 5 and 6) but could affect results for my next suggestion.

Lines 224-230: One thing you could do here to help your argument is plot the probability distribution for the relative change in pO₂ across the Ediacaran. You could then make a quantitative statement about the low likelihood of no change in pO₂. Using smoothly varying degassing forcings (see above) would make this analysis more plausible. This figure could be supplementary material if you don't have room for another figure in the main text.

Table S1 supplement. The equivalent table in the original manuscript had units for all these fluxes. Please include units as before.

Reviewers' comments and responses.

Reviewer #1 (Remarks to the Author):

The authors have made a good faith effort to address my primary criticisms in the first round of review, and although I do not think I agree on every interpretive point my criticisms of the revised manuscript would be largely either nit-picking or moving the goalposts. In particular, the sensitivity analysis is much more compelling, and the explicit incorporation of the S cycle and S isotopes into the discussion represents a significant improvement in my view.

I would be happy to see the manuscript published subject to editorial discretion.

We are glad that the reviewer finds our updated manuscript improved, and we thank the reviewer for their initial comments.

Reviewer #2 (Remarks to the Author):

This revised manuscript by Williams et al. is significantly improved and more measured with respect to their conclusions and interpretations of the modeling. The addition of the uncertainty envelopes is a much needed element to this work, and their approach using the Monte Carlo simulations seems to be a reasonable method given a similar approach is used in other recent modeling papers. This revision also includes several improvements to the explanation of the model, its parameters, and assumptions inherent in this approach, so I commend the authors for their hard work in this revision.

The authors seem to have also done an adequate job of addressing all of the reviewer's comments/concerns, as well as improving the figures and captions, which have only helped to strengthen and focus the discussion.

Overall I found a few areas in this revision that I think the authors should reconsider or change before this manuscript can be accepted for publication. Most of my comments are minor (see below), but one major issue that I am still unclear on is in regards to the balance between subduction and destruction/formation of carbonates and organic matter. I may have missed this in the discussion, and if so it might be good to highlight it again elsewhere in the text, but it seems to me that the authors invoke an increase in degassing rates as a function of subduction, but that this subduction does not increase the amount of carbonate/organic matter that is also subducted. The model requires that if CO₂ from degassing increases, that there is a net increase in C burial in equal carbonate and organic carbon proportions to balance CO₂ levels, but increasing overall atmospheric O₂ levels from higher overall organic burial rates. But what happens to these rocks? Does the model require that these rocks are formed but that they are kept from being subducted and ultimately recycled? Nowhere did I find in the text where in the ocean basin this organic burial occurs, so one has to assume that it can be equally distributed in the global ocean (I suspect most is concentrated along continental margins, but these margins cannot hold an infinite amount of sediment), but some of it most certainly occurs in the deep ocean. If so, some of this sediment would eventually be

subducted, thus diminishing the net amount of oxygen produced from organic burial. Again, unless I missed something, this I think is a critical component to this story, potentially a major flaw, and something that the authors need to address still.

The model CO₂ flux from subduction-degassing is made up of a flux from organic carbon degassing (*ocdeg*) and from carbonate carbon degassing (*ccdeg*) (equations 25 and 26 in the SI, also shown in figure 2). So buried organic carbon is assumed to eventually be subducted in our model. Table S1 shows that the reservoir of buried organic carbon (G) includes the 'ocdeg' (organic C degassing) flux as a sink, and shows that this flux is also a sink of atmospheric O₂.

The modelled increase in subduction and degassing rates results in more organic carbon being subducted, and this buffers against some of the oxygen rise that would otherwise occur. The increased burial of organic carbon that we propose also acts to increase the size of the G reservoir, which drives an increase in the subduction-degassing rate 'ocdeg'. So overall we believe the model fully satisfies the concerns of the reviewer.

We have explicitly noted in the revision that COPSE considers subduction-degassing of organic carbon.

A few minor issues:

Line 49 – Something is missing in this sentence. Perhaps change it to “concentrations”?

We have changed 'concentration' to 'concentrations'

Line 58 – A reference is needed here for the temperature estimate of Cambrian seas.

We have added a citation here.

Lines 75–109 – Please be consistent with the use of ocean-atmosphere system. For example, Lines 77 and 81 use ocean-atmosphere (which I prefer), but lines 87 and 100 are atmosphere-ocean. I am griping here only because I was once corrected on this same point and understand now that the ocean component is more important than the atmosphere.

Lines 87 and 100 have been changed to 'ocean- atmosphere' or 'ocean and atmosphere'

Line 93-94 – By “sped up”, do you mean that there was a higher flux of organic burial or increased organic burial rates?

We mean a higher flux of organic carbon burial. The text has been amended to state this explicitly.

Line 177 – Seems like it would be helpful to clarify that this oxygenation is ultimately accomplished by pyrite burial, as a function of increased sulphate reduction.

We have added this

Line 529 – Avoid the 'isotope slang' of lighter and replace with less positive.

We have changed 'lighter' to 'less positive' as suggested.

Lines 534–537 – I’m confused by this statement, which invokes euxinia and oxygenation all at once. A little clarification would help.

The redox sensitive trace elements are typically enriched under locally-euxinic conditions, but their relative concentrations over time in these sediments are thought to be reflective of changing global ocean inventories, driven by the global marine redox state, rather than local conditions. This is explained in detail in the studies we cited so the line about euxinia is removed from the figure caption in order to reduce confusion.

Line 553 – In the figure you might consider adding “ $\delta^{13}\text{C} = \#\#$ ” under each reservoir just to be crystal clear.

We have added this

Line 572 – Because every other panel has units associated with the values, you might as well add % after $\delta^{13}\text{C}_{\text{carb}}$ and $\delta^{34}\text{S}_{\text{seawater}}$.

We have added this

Line 579 – al should be “al.”

This has been changed as requested.

Line 581 – Table 1 should be capitalized.

This has been capitalized.

Reviewer #3 (Remarks to the Author):

Second Review of “A Tectonically Driven Ediacaran Oxygenation Event” by Williams, Mills, and Lenton

The revised manuscript is much improved. Most of my comments have been addressed, and the new Monte Carlo analysis is close to what I had in mind when I suggested including an uncertainty analysis in my first review. The authors’ thoughtful responses and thorough revisions are appreciated. With that said, there are still a few remaining issues that I would like to see addressed before this paper is accepted for publication:

Firstly, I don’t fully understand how reduced gas fluxes were incorporated into the model. Table 1 seems to indicate a modern range for the $\text{H}_2(?)$ flux of 0 to 2.2 Tmol ($\text{H}_2(?)$)/yr. To ensure modern day carbon fluxes balance, an equal ridge degassing flux of CO_2 was added to compensate for the reduction in oxidative weathering required to balance the oxygen cycle. This seems reasonable. However, I am unsure as to the functional form of “rgf”. I would assume that the functional form is something like $\text{rgf} = \text{modern_H}_2\text{degassing} * D$ where D is relative subduction zone length. Is this true? If so, then this should be clearly stated and explained in the supplement. If some alternative parameterization was used, then this should be justified. I don’t think it would be reasonable to sample the modern range of ridge degassing fluxes without any relationship to ridge length; the whole point of my original comment on this subject was that the H_2 flux would rise with total CO_2 degassing, partially compensating for the enhanced carbon throughput. I also don’t understand what

is meant by “sedimentary reservoirs being equated to 0 with the addition of a ridge has flux” – please clarify what this means and why it has been done.

We regret that the ‘rgf’ functional form and a brief section of explanatory text was omitted from the SI by mistake, and we have amended this in the revision. The functional form of ‘rgf’ is exactly as stated by the reviewer. As such, the flux does act in the way intended by the reviewer.

The added explanatory text includes a note that the large sedimentary reservoirs (Corg, Ccarb, pyrite and gypsum) in COPSE are now held constant for this work, which is what was alluded to in the second phrase. This modification is justified on the basis that these reservoirs do not change significantly over the relatively short timeframe considered in the paper, and is necessary because otherwise we would need to couple these reservoirs to the mantle to accommodate the ridge CO₂ flux into our present day steady state (e.g. as explored by Hayes and Waldbauer, 2006, Phil trans B). Whilst such an extension would be interesting, it would not impact the results of this paper due to the very slow changes in relative crustal carbon inventories.

We have now included this information and improved its presentation in the SI.

Second, although the Monte Carlo analysis in the revised manuscript is an improvement over the original analysis, I have a few additional recommendations. I appreciate that the uncertain parameters I highlighted in my first review have been broadly sampled (Table 1). This makes the paper’s key results much more convincing. However, the scope of the Monte Carlo analysis is somewhat limited in that only 5 parameters were varied. I suspect that these parameters are among the most important in controlling the magnitude of the oxygen rise—hence why I highlighted them in my original review—but they probably aren’t the only important parameters. The COPSE model has many other parameters and parameterizations, and uncertainty in these could affect the paper’s quantitative conclusions (Reviewer 1 clearly shares similar concerns).

The overall result that oxygen increased by ~50% is probably relatively robust to these uncertainties. However, I am less certain that the absolute size of the oxygen reservoir would remain around 0.2-0.3 if uncertainties in other parameterizations were taken into account. This is important because in the paper’s discussion on animal O₂ requirements, it is claimed that the model predicts a transition from O₂ levels below 0.25 PAL to above 0.25 PAL in the Ediacaran, which is argued to be consistent with crossing the minimum oxygen requirement threshold for the diversification of life in the Cambrian. I am concerned that if the Monte Carlo analysis was expanded to account for uncertainties in other key parameters, then oxygen levels may often remain well below (or well above) 0.25 PAL throughout the entire Ediacaran, and that the number of model runs where <0.25 PAL to >0.25 PAL transitions occurs could be rare.

To offer some concrete examples, I suspect the initial pO₂ value at 635 Ma depends on parameterizations for oxidative weathering (eq 18 in the supplement) and organic carbon burial (equations 6 and 7 in the supplement). Parameterizations of the phosphorus and sulfur cycles will similarly affect the starting pO₂ values at 625 Ma. Additionally, absolute steady state pCO₂ will depend on the temperature-dependence and CO₂-dependence of granite and basalt weathering (equations 14 and 15 in the supplement, and the climate sensitivity in equation 30). Uncertainties in these carbon cycle parameters will, in turn, affect absolute oxygen reservoirs. In some respects, the statement that pO₂ probably increased from <0.25 PAL to >0.25 PAL is equivalent to stating that pCO₂ remained bounded between 12-18 PAL throughout the Ediacaran (Fig. 5B and 5C), which is quite a strong statement that may not be justifiable with an uncertainty analysis varying only 5

parameters.

There are several ways this could be remedied. Ideally, the authors should do a more comprehensive Monte Carlo analysis to investigate the sensitivity of their results to varying a larger set of parameters. If the range in absolute oxygen levels is not strongly affected by this analysis, then the paper's conclusions would be strengthened and the implications for the Cambrian diversification would become even more compelling. Alternatively, if the authors do not wish to expand their Monte Carlo analysis, they should at least highlight the limitations of their uncertainty analysis in the main text and abstract. For example, these limitations should be noted when discussing absolute oxygen levels in the "Animal O₂ requirements" discussion.

This is an important point and we have done our best to address it in full. Firstly, we have expanded the Monte Carlo analysis to include a wider range of parameters, based on those mentioned by the reviewer (see new Table 1). Because model parameters were originally chosen towards the middle of available estimates, this further analysis does not change the mean predictions significantly but does (predictably) widen the uncertainty window. But we do agree with the key point here – that the absolute O₂ levels predicted, whilst representing a current model-based best-guess, may be shifted up or down by consideration of further model processes or parameters, including those not currently included in the model. Thus, we have made these limitations clear in the manuscript and abstract as requested. We have also added a note that the COPSE model was designed to make such quantitative predictions as well as possible, and the outputs have been tested against multiple independent lines of geochemical data over the Phanerozoic, which improves confidence in the quantitative results.

Additional comments:

Lines 94-95 and Supporting Information 1. Somewhere in here it would be helpful to summarize some version of your response to my comment (6) in my original review i.e. point out that even though there is probably a step change in relative organic burial from the mid Proterozoic to Phanerozoic, within the Neoproterozoic itself there is no strong trend.

We have added this to the manuscript.

Lines 183-185. It's not entirely clear where this linear increase in uplift comes from. Here, it makes it seem like it is derived from sedimentary records (is this Fig. 4 in reference 42?), but elsewhere (line 32 in the supplement) it seems like it is chosen to fit the Sr isotope record. The response to reviewers makes it seem like the Sr isotope record is being used, but this should be clarified in the main text.

The linear increase in uplift is chosen to satisfy the Sr isotope record, but is also guided by the sediment accumulation reconstruction of Hay et al. (indeed fig 4 in that paper). We have made this clear in the revised manuscript.

Lines 203-204: Does this mean that the degassing forcing is varying stochastically and dramatically about the median curve in Fig. 1A every 10 Myrs? Would it make more sense to use an ensemble of smoothly varying outgassing curves because stochastic variation on such short timescales is likely unphysical? This probably doesn't make too much difference for the final figures (Fig. 5 and 6) but could affect results for my next suggestion.

The reviewer is correct that the degassing forcing is varying stochastically every 10Myrs over the entire uncertainty range. The maximum variation possible under this approach is around x1.75 to subduction rate over 10 Myrs, which is a similar rate of change to that proposed by e.g. Coltice et al. (2013, EPSL) for Mesozoic crustal production. Thus we do not believe this is too extreme to be unphysical, and believe the current method gives the stiffest test of the system. Given the large ensemble size, we do not believe that this method is adversely affecting the analysis suggested in the next section.

We have updated the text to make this process clear.

Lines 224-230: One thing you could do here to help your argument is plot the probability distribution for the relative change in pO₂ across the Ediacaran. You could then make a quantitative statement about the low likelihood of no change in pO₂. Using smoothly varying degassing forcings (see above) would make this analysis more plausible. This figure could be supplementary material if you don't have room for another figure in the main text.

This is a great suggestion, and we have calculated such a probability distribution for our model output data. We present a histogram of % pO₂ change between 660-640Ma and 560-540Ma (averaged across these two time periods either side of the degassing increase) and find that over two thirds (6804 out of 10000) of our model runs display an increase in pO₂ by between 25 and 75%, whereas only 310 of our model runs display a decrease in pO₂ across this time period. This is displayed and explained in the manuscript and is a really nice addition. Again, due to the very large ensemble size we do not believe that smoothing the degassing forcing is required here.

Table S1 supplement. The equivalent table in the original manuscript had units for all these fluxes. Please include units as before.

We have added units to the table of fluxes.

REVIEWERS' COMMENTS:

Reviewer #2 (Remarks to the Author):

This revised manuscript has been significantly improved and I feel that the authors have adequately addressed the issues/suggestions made by the reviewers. My recommendation is that his manuscript be accepted for publication.

Reviewer #3 (Remarks to the Author):

My comments have been fully addressed and I am happy to recommend the paper be published in its current form. The authors should be congratulated on a thought-provoking and compelling paper.

I have just one minor comment that could be addressed in proofs: the Monte Carlo time series in Fig. 5 begin at 640 Ma, but the baseline comparison in Fig. 6 compares 660-640 Ma to 560-540 Ma. Ideally, either the Fig. 5 time series should be expanded back to 660 Ma, or the baseline for Fig. 6 should be changed to 640-620 Ma for consistency and easy comparison.

Reviewers' comments and responses.

Reviewer #2 (Remarks to the Author):

This revised manuscript has been significantly improved and I feel that the authors have adequately addressed the issues/suggestions made by the reviewers. My recommendation is that his manuscript be accepted for publication.

We are glad that the reviewer finds our updated manuscript significantly improved, and we thank the reviewer for their comments.

Reviewer #3 (Remarks to the Author):

My comments have been fully addressed and I am happy to recommend the paper be published in its current form. The authors should be congratulated on a thought-provoking and compelling paper.

We are glad that the reviewer finds our updated manuscript significantly improved, and we thank the reviewer for their comments.

I have just one minor comment that could be addressed in proofs: the Monte Carlo time series in Fig. 5 begin at 640 Ma, but the baseline comparison in Fig. 6 compares 660-640 Ma to 560-540 Ma. Ideally, either the Fig. 5 time series should be expanded back to 660 Ma, or the baseline for Fig. 6 should be changed to 640-620 Ma for consistency and easy comparison.

This is a good point, and we have changed Figure 6 in the main text and Supplementary Figure 4 such that they both compare 640-620 Ma to 560-540 Ma as suggested.